# Music listening evokes story-like visual imagery with both idiosyncratic and shared content

**Sarah Hashim**[1]*, **Lauren Stewart**[1], **Mats B. Küssner**[1,2‡], **Diana Omigie**[1‡]

**1** Department of Psychology, Goldsmiths, University of London, London, United Kingdom, **2** Department of Musicology and Media Studies, Humboldt-Universität zu Berlin, Berlin, Germany

‡ MBK and DO are shared senior authors to this work.
* shash002@gold.ac.uk

**Data Availability Statement:** All data files are available from the OSF database: https://osf.io/nf4x7/?view_only=081602e5aca94788bb959b48ed8b47ef.

## Abstract

There is growing evidence that music can induce a wide range of visual imagery. To date, however, there have been few thorough investigations into the specific content of music-induced visual imagery, and whether listeners exhibit consistency within themselves and with one another regarding their visual imagery content. We recruited an online sample (N = 353) who listened to three orchestral film music excerpts representing happy, tender, and fearful emotions. For each excerpt, listeners rated how much visual imagery they were experiencing and how vivid it was, their liking of and felt emotional intensity in response to the excerpt, and, finally, described the content of any visual imagery they may have been experiencing. Further, they completed items assessing a number of individual differences including musical training and general visual imagery ability. Of the initial sample, 254 respondents completed the survey again three weeks later. A thematic analysis of the content descriptions revealed three higher-order themes of prominent visual imagery experiences: *Storytelling* (imagined locations, characters, actions, etc.), *Associations* (emotional experiences, abstract thoughts, and memories), and *References* (origins of the visual imagery, e.g., film and TV). Although listeners demonstrated relatively low visual imagery consistency with each other, levels were higher when considering visual imagery content within individuals across timepoints. Our findings corroborate past literature regarding music's capacity to encourage narrative engagement. It, however, extends it (a) to show that such engagement is highly visual and contains other types of imagery to a lesser extent, (b) to indicate the idiosyncratic tendencies of listeners' imagery consistency, and (c) to reveal key factors influencing consistency levels (e.g., vividness of visual imagery and emotional intensity ratings in response to music). Further implications are discussed in relation to visual imagery's purported involvement in music-induced emotions and aesthetic appeal.

**Funding:** The author(s) received no specific funding for this work.

**Competing interests:** The authors have declared that no competing interests exist.

# 1. Introduction

Listening to programmatic excerpts of music, such as *Peter and the Wolf* by Sergei Prokofiev, may elicit the imagination of several different scenes. For some listeners, it might enliven childhood memories of first encounters with the piece, while for others, it may render images of the wolf, dark forests, and feelings of threat. In some cases, particular themes represented by different instruments may even give rise to the visualisation of specific characters from the tale [1, 2]. All are plausible situations afforded by listening to the piece. However, such a plurality of possibilities raises the question of the extent to which listeners are prone to the same mental experiences when listening to a piece, or whether listeners' life-long learned experiences are too complex to allow for such shared visual imagery.

For most listeners, forming a narrative in their mind's eye is a way to engage with heard music [3, 4], and such narrative sequences are often reported to be vivid and multi-thematic experiences [5, 6]. A few investigations into visual imagery content during music listening have begun to shed light on this idea of imagery consistency across listeners [7] and potential influencing factors [6]. However, the extent to which listeners exhibit similarities in their own visual imagery across listening situations is still an open question.

## 1.1. Semantic associations and what listeners imagine

Musicologists commonly speak of a 'narrative dimension' [8] to music and refer to the explicitness through which this might be communicated by drawing parallels between music and language [1, 8–10]. Research has shown that semantic associations inform our understanding and perception of music, and that the process of narrativizing music is rooted in, and requires access to, linguistic cognitive resources. A study by Koelsch et al. [11] revealed that priming words with musical stimuli induced the same electrophysiological signature (the N400) as when priming with linguistic stimuli, suggesting that as with language, listeners are capable of extracting meaning from musical stimuli. Interestingly, though, despite links between music and language processing being repeatedly demonstrated [12–16], they seem to nevertheless also be subserved by distinct cognitive mechanisms: Barraza et al. [17] used EEG to reveal that while linguistic processing appears to work on the basis of confirming participants' prior expectations for word primes, music listening implicates a post-hoc cognitive strategy, focused more on forming meaning where, due to the often subjective nature of deriving musical meaning, there is a *lack* of it. Such findings suggest that visual imagery in response to music may be a loose and unstructured process.

Given the fact that semantic associations can be formed from music, an interesting question that follows is the type of content that listeners imagine in response to music. In an online survey, Küssner and Eerola [3] asked participants to provide descriptions of the visual imagery that they typically experience while listening to music. They found that the most common type of visual imagery were natural landscapes and personal memories. Additional content included images of musical performances as well as abstract types of visual imagery such as colours and geometric shapes.

Empirical research demonstrates that people's thoughts when listening to music are generally often visual in nature, with mind wandering, or 'daydreams', tending to occur more often in the form of visual images than in the form of words [18, 19]. While examining the effects of music on thought content and valence, Koelsch et al. [19] found visual imagery to be a characteristic of mind wandering in response to music that evokes heroism and sadness. An investigation into the metaphors evoked by music carried out by Schaerlaeken et al. [20] revealed that these could be represented by five main attributes (Flow, Movement, Force, Interior, and Wandering) with visual imagery emerging as a prominent mode in which they are realised. Studying visual imagery in response to music promises theoretical insights into how cross-

modal associations [21] and meaning making emerge [22]. Taken together, these studies provide some clues into the types of visual imagery content that may be expected to be derived from music listening.

## 1.2. How stable is music-induced visual imagery?

To date, visual imagery has been assumed to be a largely idiosyncratic experience, with listeners prone to conjuring mental images not only through acoustic features, but also through processes involving past life events and contextual information [23]. Yet, the notion of consistency has been sparingly addressed in music-induced visual imagery research.

Recent related work by Margulis et al. [6] has investigated the formation of imagined narratives–mental storytelling being formed with the potential (but not necessity) to appear visually–and suggests that imagined narratives may be more widely shared than previously thought. With two sets of independent samples recruited from the US and China, they used natural language processing techniques to analyse open-text reports of narratives formed in response to instrumental music. They found that descriptions written by individuals who shared an underlying culture received higher consistency levels than reports written by those who did not.

These results offer strong support for the semantic similarities that individuals can have in their imagined narratives to music. However, it is important to note that consistency there was made in relation to general narrative formation and not with respect to visual imagery. Thus, it is possible that focusing specifically on music-induced visual imagery would yield a less pronounced similarity level.

In an influential paper, Cross [12] proposes that music possesses a *floating intentionality* whereby different listeners derive different meanings from the ongoing events in music. In line with the work of Margulis and colleagues, one might expect that individuals will tend to show higher consistency of visual imagery within themselves than with one another. However, such a possibility has never been empirically tested. One aim of the current research was therefore to shed light on this matter by comparing the degree of similarity in listeners' visual imagery reports within themselves (across two surveys) to the degree of similarity in visual imagery reports they show with other listeners.

## 1.3. Emotion and aesthetic appeal

Previous literature hints at the idea that one's experience of visual imagery may somewhat be related to their hedonic responses to music. Music-induced visual imagery has been previously associated with aesthetic and emotional engagement with music, but the evidence of such links remains limited [24, 25]. In one study, Belfi [26] investigated factors contributing to the aesthetic appeal of classical, jazz, and electronic music. Participants were required to listen to tracks taken from each genre and to report on their experience with respect to the vividness of music-induced visual imagery, arousal, emotional valence, and liking/aesthetic appeal of the music. Across genres, it was found that emotional valence and visual imagery vividness were similar in the degree of their predictive influence over musical aesthetic appeal.

In another study, Presicce and Bailes [27] established a clear association between the continuous ratings of visual imagery that participants reported in response to a selection of piano pieces and their continuous ratings of engagement with the music. Finally, in an fMRI study, Koelsch et al. [28] showed evidence of an interaction between emotion and visual areas of the brain during music listening. Interestingly, listening to fearful, compared to joyful music, led to greater interactions between the visual cortex and the superficial amygdala, perhaps due to the heightened vigilance fearful music elicits. Thus, associations between music-induced visual

imagery and emotion induction are evident, although the directionality of this relationship is still a widely debated topic supported by contrasting evidence [29, 30].

As previous studies have stated that extensive prior experience with the visual arts may be expected to influence the experiences of visual imagery that individuals have (mental imagery being regarded as a useful tool in creative processes including the visual arts, music, and dance [31, 32]), the current study also investigated how the prevalence, vividness, and consistency of music-induced imagery reported is influenced by this individual difference.

Taken together, it is apparent that visual imagery may be at least one tool with which one is 'drawn in' to music and by which music engages and induces emotions in a listener. However, there is a need to explore this with a larger sample than has been used in previous research and with a more nuanced approach that can discern particular types and subtypes of visual imagery during music listening.

## 1.4. The current research

In light of the reviewed literature, the present work sought to address four main aims:

i. To run a thematic analysis to create a hierarchical framework highlighting the prevalent codes and overarching themes found in descriptions of music-induced visual imagery content,

ii. To ascertain the extent of the consistency of visual imagery within and across individuals during music listening,

iii. To examine potential behavioural factors and individual differences (general visual imagery ability, musical training, and participation in the visual arts) that may be driving the rates of within- and across-participant consistency levels,

iv. To test the extent to which the prevalence and vividness of visual imagery is associated with the emotional intensity and aesthetic appeal of music, as well as an array of individual differences (general visual imagery ability, musical training, and participation in the visual arts).

Importantly, we anticipated that storytelling, including action-based imagery [20] (but see also [33, 34]), may emerge as a prominent form of visual imagery (H1) [3, 4, 6], and that while, in line with Cross [12], within- and across-participant consistency would be shown to be modest, there would be higher within-participant consistency than across-participant consistency (H2), due to the reduced set of factors that can lead to variance (i.e., no differences in individual listening experience or personal memories when comparing individuals with themselves, in contrast to when comparing with different individuals).

Further, we predicted that we would be able to replicate previous reports of a relationship between visual imagery and both emotion induction [30, 35–37] and aesthetic appeal [26]. We specifically predicted that the prevalence and vividness of visual imagery would both be predicted by ratings of emotional intensity (H3) and music liking (H4), in line with previous reports of positive links between these phenomena [23, 25, 26, 35].

Next, with regard to inter-individual differences, we predicted that the prevalence and vividness of visual imagery in response to music would be strongly associated with general visual imagery abilities (H5) but show a weaker relationship with musical training (H6); this is due to previous findings that musical expertise had little predictive influence over visual imagery content [5] and that while musically trained individuals may possess higher imagery abilities, this is not necessarily true of *visual* imagery [38].

Finally, we nevertheless predicted that the higher imaginative freedom of those who regularly engage with the visual arts may allow them to experience music-induced visual imagery more frequently and vividly than those who do not regularly engage with the visual arts (H7).

## 2. Methods

### 2.1. Participants

353 participants (153 female, 198 male, 2 prefer not to say) aged 18–66 years (*Mean* = 26.41, *SD* = 9.41) were recruited using either Prolific, an online participant recruitment platform, or word-of-mouth. The same sample was invited three weeks later to once more take part, with 254 participants (102 female, 149 male, 3 prefer not to say) aged 18–66 (*Mean* = 26.85, *SD* = 9.04) retaking the survey. See Table 1 for further demographic information.

This research has received ethical approval from the Ethics Committee of the Department of Psychology at Goldsmiths, University of London. Participants provided online written consent and received monetary compensation or course credit for their time.

### 2.2. Materials and stimuli

Three film music stimuli conveying happy, tender, and fearful emotions were selected from Eerola and Vuoskoski's [39] database (see, https://www.jyu.fi/hytk/fi/laitokset/mutku/en/research/projects2/past-projects/coe/materials/emotion/soundtracks for access to the original stimuli). These excerpts were obtained from the catalogue of extended 1-min film excerpts (see Appendix of [40], or see, https://www.jyu.fi/hytk/fi/laitokset/mutku/en/research/projects2/past-projects/coe/materials/emotion/soundtracks-1min for access to the original stimuli), validated to be unfamiliar to most listeners and to still convey the intended emotions even in their shorter form. In terms of the films that the tracks were taken from, the excerpt conveying happy emotions was taken from *The Untouchables* soundtrack (track 6, number 071 from Eerola and Vuoskoski's set of 110 tracks). The tender excerpt is from the *Shine* soundtrack (track 10, number 042 from set of 110 tracks). Finally, the fearful excerpt is from the *Batman Returns* soundtrack (track 5, number 011 from set of 110 tracks). In order to ensure uniformity amongst the musical excerpts, as well as to control the overall length of the survey, all excerpts were edited to last a duration of 45 seconds using Audacity (Version 2.3.2.0). These were also edited to finish with a fade-out to avoid an abrupt ending.

Visual imagery ratings were obtained using two items from Pekala's Phenomenology of Consciousness Inventory [41], a 53-item questionnaire that measures a variety of personal

**Table 1. Countries of residence of the samples from survey 1 and survey 2.**

| Country of Residence | Survey 1 (N = 353) | | Survey 2 (N = 254) | |
|---|---|---|---|---|
| | Sum | Proportion | Sum | Proportion |
| United Kingdom | 79 | 22.4% | 38 | 15.0% |
| Portugal | 59 | 16.7% | 48 | 18.9% |
| Poland | 48 | 13.6% | 37 | 14.6% |
| Italy | 18 | 5.1% | 16 | 6.3% |
| Canada | 18 | 5.1% | 14 | 5.5% |
| Mexico | 16 | 4.5% | 15 | 5.9% |
| Germany | 14 | 4.0% | 3 | 1.2% |
| Greece | 13 | 3.7% | 12 | 4.7% |
| Other | 88 | 24.9% | 71 | 28.0% |

perceptual experiences revolving around consciousness. Participants were asked about the prevalence of visual imagery in their experience (1 = I experienced no visual imagery at all, to 7 = I experienced a great deal of visual imagery), and the vividness of their imagery (1 = My visual imagery was so vague and diffuse, it was hard to get an image of anything, to 7 = My visual imagery was so vivid and three-dimensional, it seemed real).

The content of visual imagery was measured using an open-text question. Participants were asked to describe the content of their visual imagery (if any) with no limit to the length of their descriptions required. Liking ratings in response to the music were measured using a 5-point Likert scale, from 1 = Dislike a great deal, to 5 = Like a great deal. Emotional intensity ratings were also measured using a 5-point Likert scale, from 1 = Not at all intense to 5 = Extremely intense. Participants were also asked to report on the musical aspects they thought contributed to their visual imagery, but data from this question will be addressed elsewhere.

Finally, we measured three aspects of individual differences. The Musical Training dimension of the Goldsmiths Musical Sophistication Index (Gold-MSI) [42] was administered in order to gauge the extent of individuals' musical experience. The Vividness of Visual Imagery Questionnaire (VVIQ) [43], which comprises 16 statements to which individuals are instructed to form a visual mental image in their minds, was administered as an independent measurement of visual imagery ability, along a 5-point Likert scale from 1 = No image at all, you only 'know' that you are thinking of an object to 5 = Perfectly clear and vivid as real seeing. Finally, any experience with activities associated with the visual arts ('*Do you participate in any activities associated with the visual arts*?') was also probed. This was a binary yes/no question, with an option to elaborate on the type of activity experienced if answered in the affirmative.

## 2.3. Procedure

Participants were first provided with the aims, instructions, and a definition of visual imagery as *"the spontaneous formation of visual images or pictures in your mind's eye. Your imagery experience is completely subjective, and it is completely acceptable not to have experienced any imagery at all"*. Survey 1 took approximately 12 minutes to complete. The presentation order of musical stimuli was randomised across participants.

Participants were advised to use good-quality headphones and to have minimal outside disturbance throughout the study. For the three main trials, participants were first presented with the musical excerpt. They were instructed to listen to the whole excerpt and told that on the next page they would be presented with questions regarding their experience of the music. They were advised to pay attention to any visual imagery that they may be experiencing, and the musical characteristics that may have contributed to the imagery (question addressed elsewhere), as they listened. On a first response page, participants were asked to rate the prevalence (the amount experienced) and vividness (the clarity with which it was experienced) of their visual imagery. These ratings were followed by the open-text question asking them to describe the content of their visual imagery (if any). Finally, participants were asked to rate how much they liked the music, and how intense any felt emotional response to the music was.

After this, participants completed the VVIQ, the musical training dimension of the Gold-MSI, then indicated whether they have participated in any activities associated with the visual arts.

Three weeks later, participants completed an almost identical survey, excluding the VVIQ, Gold-MSI, and question about their experience with the visual arts. Certain demographic questions were collected once again, such as an anonymous ID, but also age, and nationality, to improve the chances of confidently being able to match participant data across both surveys. This second survey took approx. 8 minutes to complete.

## 3. Analysis

### 3.1. Thematic analysis of open-text reports of music-induced visual imagery content

Using Braun and Clarke's [44] approach, we implemented a thematic analysis on the responses to the open-text question ('*Please describe the content of your visual imagery (if any)*') to identify the prominent themes that emerged in terms of visual imagery in response to the three musical excerpts. This thematic analysis was carried out using Microsoft Excel (Version 16.43) while further quantitative analyses were calculated using R (Version 4.2.3) [45].

The thematic analysis was carried out by two independent coders to reduce the risk of personal bias. The dataset was analysed as one large unit independent of excerpt emotion with the aim of producing a rich account of each developing theme. In general, to overcome any potential variability in language as a result of participants' wide range of personal and musical backgrounds, we discerned meaning from the data at an explicit level (i.e., a literal interpretation of the text).

The dataset from survey 1 comprised a total of 1,059 reports across the three musical excerpts, while survey 2 comprised a total of 762 reports. Two coders independently examined the dataset and identified Level 3 codes that could encompass key aspects of each description, as well as making note of as many ideas and patterns that may benefit the final model as relevant. Commonalities between specific terms were identified from Level 3 (L3) codes, which were sorted and categorised into distinct groups comprising Level 2 (L2) codes. The suitability of the subthemes in this level was reviewed by the two coders, including whether to collapse redundant subthemes or to divide ones that were too diverse. The L2 codes were finally combined to form higher-order Level 1 (L1) themes. The final structure was discussed by the two coders, confirming that it formed an effective hierarchy that offered a parsimonious overview of the content of the free-form descriptions.

### 3.2. Measuring consistency within and across reports

We used the Jaccard coefficient index, a conservative measure of agreement that enables assessment of the level of overlap between any two lists (0 = no similarity at all, and 1 = perfect similarity):

$$J(A, B) = \frac{A \cap B}{A \cup B}$$

In order to assess consistency, we needed a way of referring to each visual imagery code in a systematic way. Thus, we created a labelling system where all theme levels from our thematic analysis were assigned a numeric label as a unique identifier. To test the efficacy of these labels as a way to sort each participant report, the two coders first used this system to independently label a subset of 60 participant reports (5–6% of total) from the dataset, using L3 codes (the most detailed level of our visual imagery codes). Each individual report was assigned labels pertaining to the content of their descriptions. This first attempt by the coders yielded a Jaccard similarity average of 69% between Coders 1 and 2. Coders discussed any comments or issues that resulted from this effort, including minor reorganisation of the code levels. As a final check, a new subset of 60 reports was analysed, resulting in a version that was a more feasible and intuitive organisation of the themes. The codebook was reviewed and discussed once more by the research team to address any remaining structural issues. Finally, Coder 1 used the labelling system to assign labels to the rest of the datasets for survey 1 and survey 2. The

original and coded framework can be found through the Open Science Framework using the following link: https://osf.io/nf4x7/?view_only=081602e5aca94788bb959b48ed8b47ef.

We assessed two types of consistency for each musical excerpt separately: within- and across-participant consistency. Within-participant consistency was computed by comparing the lists of themes emerging from each participant's reports across surveys 1 and 2, and across-participant consistency was computed by comparing the list of themes from each individual participant with the lists from every other individual in the sample, resulting in 352 values per participant for each musical excerpt type (leading to three groups of values) that were then averaged to create a single consistency value per individual per excerpt. With regard to across-participant consistency specifically, individuals were always compared against other individuals' responses that were within the same excerpt type.

In order to retain a precise estimation of the visual imagery types present in the data and its consistency within the sample, the descriptions of those who reported not experiencing any visual imagery or only vague imagery in one or both timepoints were excluded before running the Jaccard analyses. However, those who were coded as having no or vague imagery *were* included if they also reported additional coded experiences that could be used to compute their consistency (i.e., reports describing no or blurred visual imagery, but references to other relevant experiences such as emotional reactions or other imagery types). We also ran the same analyses on our codes only depicting visual experiences (i.e., those under the *Storytelling* higher-order theme, see section 4.1). This was to fully address one of our main aims and research questions regarding the consistency of content of visual imagery more specifically.

### 3.3. Comparing consistency levels and determining the behavioural measures and individual differences that influence them

We assessed the differences in Jaccard coefficient scores between within- and across-participant consistency within each code level (L2 (more broad) versus L3 (more detailed) codes) using linear mixed effects models. The purpose of this was to examine whether consistency levels differ when comparing within and across individuals. To test this, two models were defined, using the *lme4* [46] and *lmerTest* [47] packages in R, one including L3 consistency values as the dependent variable and the other including L2 values as the dependent variable. In both models, Consistency Type was entered as a categorical fixed effect representing whether the consistency values originate from the Within-Participant or Across-Participant analysis, with participant and musical excerpt entered as random effects in both models.

The same method was used in a subsequent stage of the analysis to test the strength of any relationships between visual imagery consistency values (within- and across-participants) and data on the behavioural experiences of music as well as individual differences. Four models were run, each including within- and across-participant consistency values for each code level (L2 and L3) as dependent variables. In each model, prevalence and vividness of visual imagery, music liking, emotional intensity, VVIQ, musical training, and visual arts participation (*yes* or *no*) were entered as fixed effects, with participant and musical excerpt included as random effects.

### 3.4. Tests of associations and differences between behavioural measures and individual differences

Finally, linear mixed effects models and t-tests were run to ascertain the strength of associations and differences with regard to behavioural ratings (prevalence and vividness of visual imagery, music liking, and emotional intensity) and data on individual differences (VVIQ, musical training, and visual arts participation). To this end, two models were run, one with

prevalence as dependent variable and the other with vividness as dependent variable, with music liking, emotional intensity, VVIQ, and musical training included as fixed effects, and participant and musical excerpt as random effects in both models. Further, independent samples t-tests were run to assess differences between those who do and do not participate in the visual arts in terms of their prevalence and vividness of visual imagery ratings.

## 4. Results

### 4.1. Thematic analysis of music-induced visual imagery

As can be observed in Table 2, the thematic analysis conducted on descriptions of visual imagery content resulted in three higher-order themes pertaining to the most prevalent topics in participant reports. As will be observed below, participant descriptions did not always include experiences that were strictly visual in content and often included descriptions related to affect, musical features, and other forms of mental imagery. This fact has been taken into consideration with regard to analyses and conclusions drawn.

A summary of the frequencies of each L2 subtheme is compiled in Fig 1 (see Fig A in S2 File for frequencies of L2 codes across music excerpts, and Fig B in S2 File for frequencies of L3 codes). We summarise each higher-order theme and their subthemes (in descending order of prevalence) as follows:

**4.1.1. Storytelling.** *Storytelling*, our first higher-order theme, is characterised by descriptions of narratives involving situations, actions, people, and locations, thus confirming Hypothesis 1. In most cases, descriptions referred to fictional situations involving non-existent

**Table 2. Thematic framework of prominent higher-order themes (L1) of music-induced visual imagery content (with L2 and L3 codes in descending order of prevalence).** Values included beside L1 codes represent their proportions within the entire framework, whereas the two values beside L2 codes reflect their proportions within their higher-order theme (left) and within the entire framework (right).

| Level 1 | Storytelling (74.3%) | | | Associations (6.4%) | | | References (19.3%) | | |
|---|---|---|---|---|---|---|---|---|---|
| Level 2 | Setting & Location (38.0%, 28.2%) | Characters (27.5%, 20.4%) | Action (17.7%, 13.1%) | Narrative (16.8%, 12.5%) | Abstract & Other (59.9%, 3.8%) | Feelings & Atmospheres (29.6%, 1.9%) | Memories (10.9%, 0.7%) | Other Media (89.5%, 17.2%) | Music (10.5%, 2.0%) |
| Level 3 | Building | Imagery of Self | Communicative Gestures | Plot Lines | Physical Reactions | Emotional Reaction/Feelings | Autobiographical | Medium | Genre |
| | Urban | Individuals | Idle/Passive | Hardship | Other Senses | Atmosphere/Mood | Episodic | Genres | Composer |
| | Nature | Royal | Mental | Conflict & Resolution | Cross-Modal | | | Specific Movie/Game/Show | Characteristics |
| | Interior & Exterior Features | Heroic | Interaction | Celebratory | | | | Features | Instruments |
| | Existing Locations | Evildoer | Musical | Inquisitive | | | | | |
| | Imagined Setting | Civilian | Leisure | | | | | | |
| | Time References | (Social) Groups | Gait | | | | | | |
| | Season | Musical | | | | | | | |
| | Weather | Animals/Insects | | | | | | | |
| | Other Objects | Characteristics | | | | | | | |

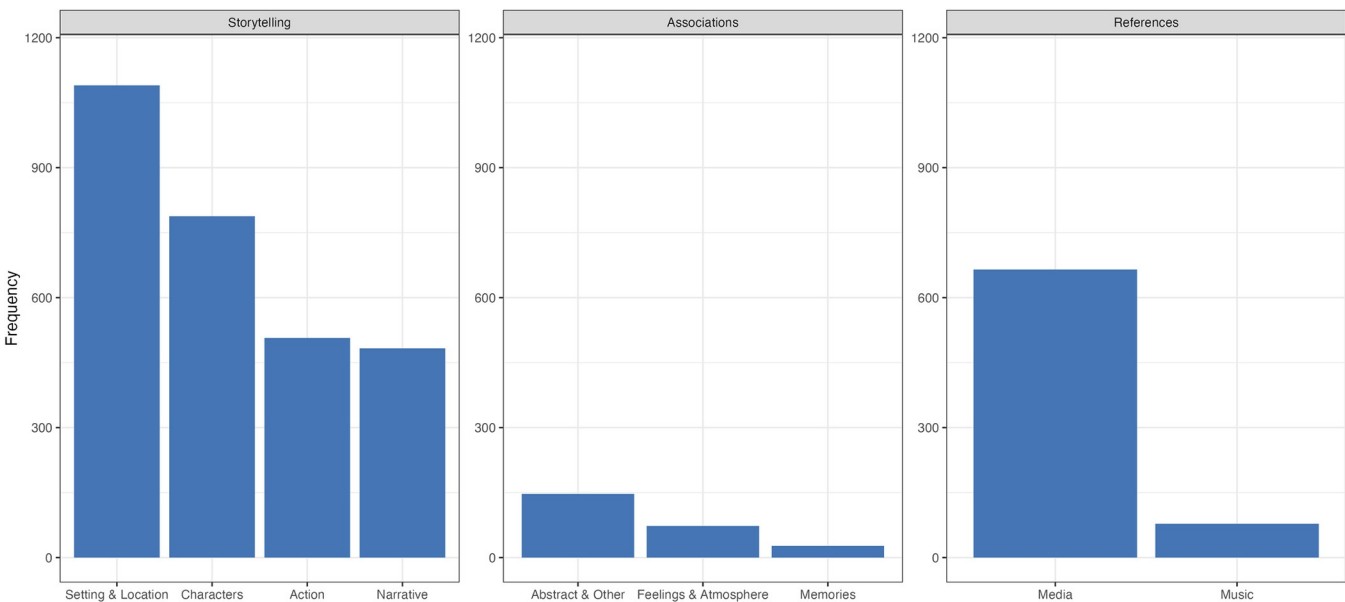

**Fig 1. Occurrence of L2 visual imagery content.**

characters, although participants sometimes also visualised themselves or other familiar individuals in imagined circumstances. This higher-order theme most explicitly encompasses details that were visualised. This higher-order theme was referred to in 74.3% of the total sample (2,868 times) throughout reports, and comprised four L2 subthemes:

i. *Setting & Location*: describing the locations, interior/exterior, and temporal (and other) details in which the scenes took place. This comprised 38.0% of the *Storytelling* theme (28.2% of total, 1,090 times, e.g., "*It was a **tree on top of a hill**", "I was in a **museum** on **a day with beautiful weather***")

ii. *Character*: referring to the presence of an individual, groups of people, animal, or imagery of oneself within content descriptions. This comprised 27.5% of the *Storytelling* theme, 20.4% of total, 788 times). This subtheme comprised (fictional or real) individuals ranging in background or societal status and includes a lower-level theme describing a range of bodily characteristics (e.g., "*I feel like I'm at the coronation of a future **king**", "**A man** in a **button-down, sleeves** rolled up to mid-arms, sitting in an apartment*")

iii. *Action*: referring to actions or activities performed by individuals contained in the descriptions. This comprised 17.7% of the *Storytelling* theme (13.1% of total, 507 times, e.g., "*A couple **walking** on the beach*", "*Man and woman **dancing** slowly*")

iv. *Narratives*: referring to plot transitions and prevalent themes portrayed by the content. This comprised 16.8% of the *Storytelling* theme (12.5% of total, 483 times, e.g., themes of celebration or conflict, "*A celebration **of a great event** with lots of people...*", "*Some scary monsters, and **very dynamic action**...*", "***At first,** I sort of had an imagery of a battlefield... **then later on** it was more sort of fairy-tale like*")

**4.1.2. Associations.** *Associations*, our second higher-order theme, occurred 6.4% of the total number of reports (247 times throughout reports). This theme encompasses a mixture of perceptual or sensorial experiences within participants' reports to the music and/or those

contained in narrative descriptions of the music, as well as some mention of abstract forms of visual imagery. In all, this higher order theme groups together multimodal and affective experiences contained within the musical experience. It comprised three L2 subthemes:

i. *Abstract & Other*: referring mostly to abstract forms of visual imagery, mental imagery additional to visual imagery, as well as physical experiences in relation to the music (e.g., "*At first, I **saw light and bright colour**s and later. . . clouds above **a red colour***", "*I felt my **eyes tremble***"). This subtheme occurred 59.5% of the *Association* theme (3.8% of total, 147 times)

ii. *Feelings & Atmosphere*: comprising 29.6% of the *Associations* theme (1.9% of total, 73 times). This subtheme describes any emotional reactions or moods portrayed by the scene or any felt by characters within the narrative or participant themselves (e.g., "***Love and happiness** in general*", "*A **scary mood**, **oppressive**, a **danger***"). Whilst not representing visual imagery, its presence within reports was considerably prevalent and constituted a prominent attribute of descriptions of visual imagery experience

iii. *Memories*: describing two types of memories that participants may have recalled during their listening experience: autobiographical (e.g., "*It **reminds** me of the old cartoons*") and episodic (e.g., "*Walking to the stage **on my graduation***"). This theme was the least prevalent amongst the other elements in the framework, occurring only 10.9% of the already very small *Associations* theme (0.7% of total, 27 times, e.g., "***My** ballet classes*")

**4.1.3. References.** *References*, the final higher-order theme, occurred in 19.3% of the total sample (743 times) throughout reports. It is comprised of two L2 subthemes and contains codes referring to details pertaining to the origin of the participant's visual imagery, and generally also includes a mixture of semantic associations (e.g., the music being characteristic of a particular genre) and/or visualised details (e.g., specific instruments or composers). It comprised two L2 subthemes:

i. *Media*: the first and majority subtheme (89.5% of the theme, 17.2% of total, 665 times), comprised references to different types of media (e.g., television, film), references to media genres (e.g., horror, action, film noir), and mentions of any pre-existing media (e.g., "***Animated Disney-style movie***", "*A **40s horror movie***")

ii. *Music*: comprised references to musical genres (10.5% of the theme, 2.0% of total, 78 times), composers, or instruments that were visualised or described as contributing to the visual imagery (e.g., "*As if I were sitting with tea in my hand next to a famous composer like **Fryderyk Chopin***", "***When the piano started playing** I imagined a dark haired male pianist on a black **piano**. . .*")

## 4.2. Do respondents show higher consistency with themselves than others?

Table 3 provides examples of reports at different (full framework) within-participant consistency boundaries. While reports at 0% reflect no discernible overlap, the 20–50% consistency range reflects minor content overlap in reports between timepoints, the 50–99% consistency range denotes significant content overlap between timepoints, and 100% reflects complete overlap and almost identical responses between timepoints.

Fig 2 illustrates the distributions of within- and across-participant consistency (and Fig C in S2 File demonstrates these distributions as a function of music excerpt type). The graphs display marked differences in range, with the within-participant distribution spanning a wider set of consistency values and additionally revealing a higher proportion of zero levels of

**Table 3. Example visual imagery excerpts at different bands of within-participant consistency L3 codes.**

| Survey 1 Excerpts | Survey 1 Codes | Survey 2 Excerpts | Survey 2 Codes | Consistency Level | Reports at this level (% of N) |
|---|---|---|---|---|---|
| *Akin to the end of a children's fantasy movie* | 1.1.1, 2.3.1, 2.3.2, 1.4.7 | *Gathering of people for a celebration of royalty* | 1.1.4, 1.4.7, 1.4.3 | 0–19.99% | 399 (55.4%) |
| *I'm watching a theatre play, and it's starting.* | 1.4.1, 1.2.5, 1.1.5 | *I have imagined that I'm in a big forest to explore it* | 1.3.1, 1.1.3, 1.2.6 | | |
| *I imagined a man playing a piano in an empty room* | 1.4.2, 1.2.5, 4.1.4, 1.3.1, 1.3.4 | *I felt like I was watching someone playing on a piano in a massive room. Beautiful, relaxing. . .and sort of nostalgic.* | 1.2.5, 1.4.2, 3.2.4, 1.3.1, 2.3.1 | 20–49.99% | 208 (28.9%) |
| *The celebration of a victory at the end of a war* | 1.1.3, 1.1.4 | *A hero triumphant return to its home village* | 1.4.4, 1.1.3, 1.3.1 | | |
| *WWI trenches, then galloping horses, then marching soldiers* | 1.1.1, 1.1.3, 1.2.6, 1.4.9, 1.2.7, 1.4.3 | *WWI, American civil war battles, armies marching* | 1.1.3, 1.4.3, 1.2.7 | 50–99.99% | 90 (12.5%) |
| *I feel like I'm at the coronation of a future king* | 1.4.1, 1.1.4, 1.4.3 | *I have the impression that I was at the coronation of the king / queen, or at some state event* | 1.4.1, 1.4.3, 1.3.1, 1.1.4 | | |
| *Lonely day at bar/cafeteria. Raining outside. Lonely afternoon.* | 1.3.1, 2.3.1, 1.3.9, 1.3.7 | *Lonely evening at the bar. Raining outside.* | 2.3.1, 1.3.7, 1.3.1, 1.3.9 | 100% | 23 (3.2%) |
| *I imagine someone of royalty getting married like for example Prince William and Kate.* | 1.4.3, 1.1.4 | *I imagine a wedding of someone of the royalty.* | 1.1.4, 1.4.3 | | |

N = 720

consistency, while across-participant consistency appears to span a much narrower value range and peaks at a relatively low level (see also Tables C and D in S1 File for precise consistency values for within- and across-participant profiles, respectively, at different Jaccard percentage ranges).

In order to test Hypothesis 2, we asked whether there were differences in consistency when comparing an individual with themselves at a later stage, on the one hand, to when comparing a given individual with the remaining sample. The model comparing Jaccard coefficient values for L2 codes between the within- (*Mean = 27.4%, Median = 25%*) or across-participant

**(A)**

**(B)**

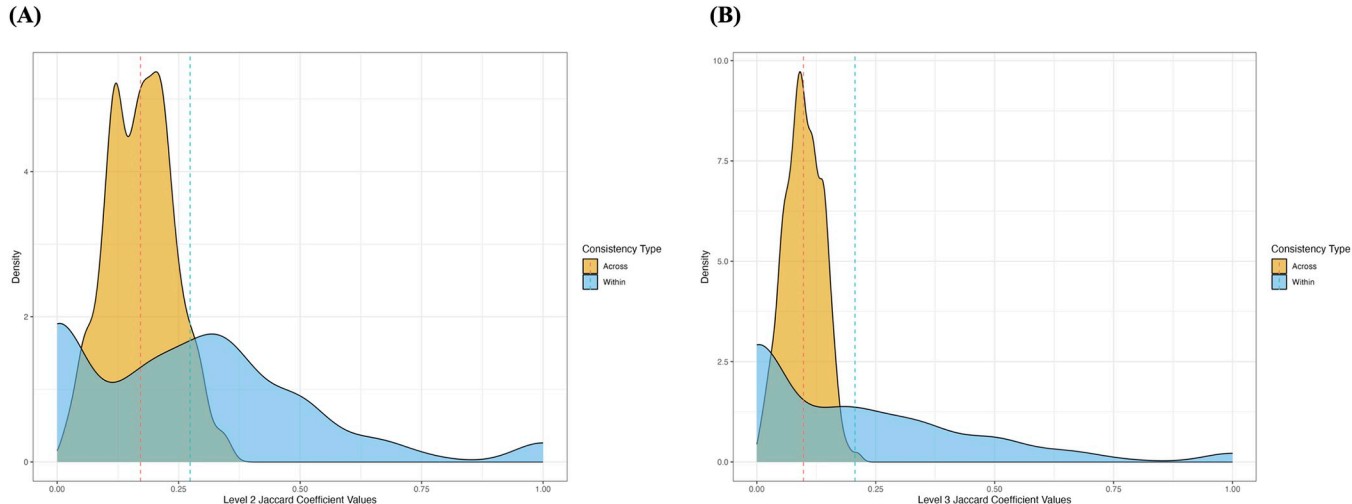

**Fig 2. Comparing density distributions of consistency values of within- (across listening situations) and across-participant (across listeners) groups with mean intercepts.** (A) Level 2 codes. (B) Level 3 codes.

(*Mean* = 17.1%, *Median* = 17.2%) distributions, indeed, confirmed there to be distinguishable differences between the two groups (ß = 0.10, SE = 0.01, *t* = 13.00, *p* < 0.001), with the within-participant group displaying overall greater consistency values than the across-participant group. Additionally, the model predicting coefficient values at the higher granularity L3 codes similarly showed there to be overall differences between the within- (*Mean* = 20.6%, *Median* = 14.3%) and across-participant (*Mean* = 9.8%, *Median* = 9.6%) groups (ß = 0.11, SE = 0.01, *t* = 14.24, *p* < 0.001).

Fig D in S2 File presents within- and across-participant distributions using only *Storytelling* codes, since these, unlike the *Associations* and *References* codes, constitute codes that characterise purely visual imagery experiences. We once more show the two distributions to possess distinct shapes and peaks as previously described for all codes. Again, the model comparing Jaccard coefficient values for L2 codes between the within- (*Mean* = 33.6%, *Median* = 28.6%) or across-participant (*Mean* = 21.5%, *Median* = 21.2%) distributions for *Storytelling* codes confirmed differences between the two groups (ß = 0.12, SE = 0.01, *t* = 12.06, *p* < 0.001). Finally, the model predicting consistency values at the higher granularity L3 codes also demonstrated overall differences between the within- (*Mean* = 26.1%, *Median* = 16.6%) and across-participants (*Mean* = 11.9%, *Median* = 11.3%) groups (ß = 0.14, SE = 0.01, *t* = 14.35, *p* < 0.001).

## 4.3. Associations between visual imagery consistency and musical experience

In the next set of analyses, we asked to what extent across-participant consistency (a participant compared with everyone else in the sample) or within-participant consistency (a participant compared with themselves) were associated with participants' experience of the music (visual imagery prevalence and vividness, liking, and emotional intensity) on the one hand, but also individual differences in general visual imagery ability, musical training, and participation in the visual arts on the other (see Table 4 for a full summary of the model results).

The model predicting L2 across-participant consistency revealed prevalence (ß = 0.01, SE = 0.00, *t* = 2.96, *p* = 0.003) and vividness (ß = 0.00, SE = 0.00, *t* = 2.49, *p* = 0.013) of visual imagery to be significant predictors of this level of consistency, whereas the model predicting L3 across-participant consistency showed only prevalence to be a significant predictor (ß = 0.00, SE = 0.00, *t* = 2.30, *p* = 0.022). No other behavioural measures or individual differences significantly predicted across-participant consistency in either level.

**Table 4. Fixed effect estimates of behavioural and individual difference measures predicting within- and across-participant consistency.**

| | Across Consistency L2 | | | | Across Consistency L3 | | | | Within Consistency L2 | | | | Within Consistency L3 | | | |
|---|---|---|---|---|---|---|---|---|---|---|---|---|---|---|---|---|
| | ß | SE | t | p | ß | SE | t | p | ß | SE | t | p | ß | SE | t | p |
| **Intercept** | 0.11 | 0.02 | 6.32 | **< 0.001***** | 0.09 | 0.01 | 7.84 | **< 0.001***** | 0.09 | 0.06 | 1.40 | 0.163 | 0.10 | 0.06 | 1.61 | 0.110 |
| **Imagery Prevalence** | 0.01 | 0.00 | 2.96 | **0.003**** | 0.00 | 0.00 | 2.30 | **0.022*** | 0.01 | 0.01 | 1.18 | 0.238 | 0.01 | 0.01 | 1.20 | 0.230 |
| **Imagery Vividness** | 0.00 | 0.00 | 2.49 | **0.013*** | 0.00 | 0.00 | 1.24 | 0.214 | 0.01 | 0.01 | 1.42 | 0.155 | 0.02 | 0.01 | 2.17 | **0.030*** |
| **Music Liking** | 0.00 | 0.00 | 0.62 | 0.534 | −0.00 | 0.00 | −0.16 | 0.869 | 0.01 | 0.01 | 0.94 | 0.356 | −0.01 | 0.01 | −1.30 | 0.193 |
| **Emotional Intensity** | 0.00 | 0.00 | 1.75 | 0.080 | −0.00 | 0.00 | −0.62 | 0.534 | 0.01 | 0.01 | 1.10 | 0.274 | 0.01 | 0.01 | 0.77 | 0.440 |
| **VVIQ** | 0.00 | 0.00 | 0.44 | 0.658 | −0.00 | 0.00 | −0.68 | 0.495 | 0.01 | 0.01 | 0.47 | 0.637 | −0.01 | 0.01 | −0.41 | 0.684 |
| **Musical Training** | −0.00 | 0.00 | −1.41 | 0.161 | −0.00 | 0.00 | −0.56 | 0.576 | −0.00 | 0.01 | −0.30 | 0.768 | 0.00 | 0.01 | 0.62 | 0.534 |
| **Visual Arts** | 0.00 | 0.01 | 0.72 | 0.470 | 0.00 | 0.00 | 0.36 | 0.716 | −0.03 | 0.02 | −1.27 | 0.204 | −0.01 | 0.02 | −0.50 | 0.618 |

*** < 0.001

** < 0.01

* < 0.05. Abbreviations: L2 = Level 2, L3 = Level 3.

Conversely, the model predicting L2 within-participant consistency revealed that none of the behavioural measures possessed any predictive influence over this consistency level. However, the model predicting L3 within-participant consistency showed a significant influence of the vividness of visual imagery (ß = 0.02, SE = 0.01, $t$ = 2.18, $p$ = 0.030). No other behavioural measures or individual differences significantly predicted within-participant consistency in either level.

## 4.4. Prevalence and vividness of music-induced visual imagery and links between visual imagery, emotional intensity, liking, and individual differences

Table 5 presents a summary of visual imagery prevalence and vividness ratings divided and aggregated by musical excerpt. These values demonstrate high proportions of visual imagery prevalence and vividness across all musical excerpts, that persist even when one considers the individual excerpt types. Higher prevalence and vividness levels of visual imagery can be seen in response to the Fearful excerpt than the Happy and Tender excerpts, which both exhibit almost equal proportions.

Confirming Hypotheses 3, 4, 5, and 6, the model predicting prevalence of visual imagery demonstrates that emotional intensity (ß = 0.70, SE = 0.04, $t$ = 15.83, $p < 0.001$), music liking (ß = 0.25, SE = 0.04, $t$ = 5.53, $p < 0.001$), and the VVIQ (ß = 0.48, SE = 0.08, $t$ = 5.95, $p < 0.001$) were all highly significant predictors. Contrastingly, musical training (ß = 0.02, SE = 0.04, $t$ = 0.65, $p$ = 0.517) did not significantly predict visual imagery prevalence.

Similarly confirming our hypotheses, the model predicting vividness of visual imagery similarly showed emotional intensity (ß = 0.70, SE = 0.04, $t$ = 15.25, $p < 0.001$), music liking (ß = 0.21, SE = 0.04, $t$ = 4.64, $p < 0.001$), and the VVIQ (ß = 0.54, SE = 0.08, $t$ = 6.45, $p < 0.001$) to be highly significant predictors, whereas, once again, musical training was not (ß = −0.02, SE = 0.04, $t$ = −0.45, $p$ = 0.655).

In response to the question regarding experience with activities in the visual arts (VA), 20.1% (n = 71) reported that they participate in activities associated with the visual arts, which included activities such as painting, photography, and graphic design. With regard to the averaged excerpt ratings between those who do and do not (NVA) participate in the visual arts, independent samples t-tests showed that those who participate in the visual arts reported significantly more visual imagery prevalence (*Mean*-VA = 4.63, *Mean*-NVA = 3.99, $t(119.4)$ = 3.78, $p < 0.001$) and more vividness (*Mean*-VA = 4.30, *Mean*-NVA = 3.74, $t(124.1)$ = 3.37, $p < 0.001$) than those who do not, supporting Hypothesis 7. There were also significant differences with regard to each of the individual musical tracks, with individuals who do take part in the visual arts reporting higher prevalence and vividness than those who do not (see Tables A and B in S1 File for full results).

**Table 5. Proportions (and frequencies) of the prevalence and vividness of visual imagery for and across the musical excerpts.**

|  | Happy | | Tender | | Fearful | | Overall | |
|---|---|---|---|---|---|---|---|---|
|  | **No VI** | **At least mild VI** | **No VI** | **At least mild VI** | **No VI** | **At least mild VI** | **No VI** | **At least mild VI** |
| **Prevalence** | 8.8% (31) | 90.7% (320) | 9.6% (34) | 90.4% (319) | 5.1% (18) | 94.6% (334) | 2.5% (9) | 97.5% (344) |
| **Vividness** | 10.9% (38) | 89% (314) | 13.6% (48) | 86.1% (304) | 5.9% (21) | 93.2% (329) | 5.4% (19) | 94.6% (334) |

Abbreviations: VI = visual imagery. "No VI" pertains to individuals who provided the lowest rating, whereas "At least mild VI" refers to ratings of 2 and upwards.

Values in brackets indicate the sum of reports within that category. Combined sum of reports for or across excerpt types for each rating that do not equate to the total number of participants (N = 353) are due to missing ratings data.

## 5. Discussion

The aims of the current study were multi-fold. Our main aim was to derive the most prominent themes present in listeners' descriptions of their music-induced visual imagery and to investigate the extent to which listeners exhibited consistency in their visual imagery reports within themselves (within-participants) and with the whole cohort (across-participants). Further to this aim, we sought to then explore whether the prevalence and vividness of visual imagery, music liking and emotional intensity, as well as individual differences in general visual imagery ability, musical training, and participation in the visual arts may be associated with patterns of consistency levels shown. However, other important aims were to replicate previous findings of a link between visual imagery and emotion [30, 37], aesthetic appeal [26], and to explore the influence of general visual imagery ability, musical training, and participation in the visual arts on music-induced visual imagery experience.

In sum, we found (a) that storytelling is the most common form of visual imagery during music listening, (b) individuals are more consistent with themselves (in terms of their visual imagery content) than when compared to other listeners, (c) evidence for all-round modest consistency levels, (d) confirmation of links between visual imagery, emotional intensity, and aesthetic appeal, and (e) evidence for a nuanced role of individual differences on music-induced visual imagery experience in terms of prevalence, vividness, and consistency. Further to these findings, we ascertained that visual imagery experience averaged across the three music stimuli was very prevalent in our sample with about 97.5% of participants reporting experiencing at least mild levels of visual imagery, and about 94.6% reporting their visual imagery to be at least a mildly vivid experience (similar proportions were also found across the individual music excerpts).

### 5.1. Storytelling is the most common form of visual imagery during music listening

The thematic analysis of our open-text question revealed that (in support of H1) story-making was a very pervasive aspect of visual imagery experience. Indeed, our first higher-order theme *Storytelling* encompassed descriptions of visual imagery ranging from locations to individual characters and was present in over 74% of reports. This finding is in line with the idea that individuals are prone to imagining narratives in response to music [4, 6], and our findings support the notion that this may significantly occur in the visual domain. In addition to showing that visualising story-making is a prevalent aspect of music listening, we were also able to offer insights into the relative prevalence of certain types of visual imagery such as the prominence of details pertaining to setting and location, followed by characters, and the actions they carried out.

We coded two further themes that accompanied our sample's visual imagery reports, *Associations*, comprising abstract visual imagery, emotion, and memories, and *References*, comprising codes regarding media comparisons, instruments, and composers. Codes underlying these themes were present in over 10% of reports, and support past arguments that music listening is a multi-modal experience [48, 49] that involves not just visual experiences but also physical, affective, and semantic connotations. In other words, in addition to reports suggesting that certain visual imagery was in fact 'seen', descriptions were also often accompanied by remarks of cross-modal aspects spanning the remaining senses, aesthetic evaluations, and music's emotional power. These findings highlight the considerable prevalence of semantic associations found within listeners' descriptions in response to the music, even when explicitly instructed to focus their attentions on visual imagery. Such patterns indicate that narrowing one's research aims on just the visual components of a listeners' experience could lead to

overestimations of its occurrence as well as overshadows its potential unique links to other semantic associations formed in response to the music; one recent study by Cespedes-Guevara and Dibben [50] showed that a considerable portion of listeners' reports of what went through their minds while listening comprised an array of semantic, personal, as well as visual experiences.

## 5.2. Participants show all-round low consistency levels but greater levels for within- than across-participants

We investigated visual imagery consistency using the Jaccard coefficient index, a measure of overlap between two lists. First, we assessed within-participants consistency by comparing each individuals' content from surveys that were administered three weeks apart. Here, we found that, overall, consistency levels tended to cluster around 20%, indicating that listeners tended to refer to almost a fifth of the same visual imagery features in both instances. Our analysis of across-participant consistency yielded similarly low results when comparing individual participants with the rest of the sample: approx. 7–13% for L3 codes and 13–22% for L2 codes.

At first glance, our findings show notable differences with a recent investigation by Margulis et al. [6]. The authors reported high levels of consistency in their sample with regard to musical narrative engagement that they suggest was, for the most part, determined by cultural experience. However, it is important to point out both the differences in what was being reported on (visual imagery vs. imagined narrative) and how consistency is estimated in the two papers. Indeed, we opted to use the Jaccard coefficient index on lists of themes, a method which takes the presence of individual cases into consideration. In contrast, Margulis and colleagues' approach focuses on the respondents' text as a whole, using a cosine similarity as a weighted index to assess semantic 'closeness' between portions of text based on their orientations on a multi-dimensional space. Approaches used by other studies to address similar questions have also included frequency-based approaches [7] (i.e., summing occurrences of certain content). Critically, the current research aimed to assess consistency with a focus on the presence or absence of particular themes and topics. By allowing analysis of consistency on the basis of themes and topics, our approach provides a way to estimate consistency levels with a focus on particular aspects of content. However, it would be beneficial to consider the advantages of applying a weighted approach (as Margulis and colleagues have done) when scrutinising visual imagery content in combination with the current study's methods of assessing discrete overlap. Incorporating both topic overlap as well as weighted similarity might provide insight on the significance of specific types of cross-modal content evidently present (as is seen within our thematic framework) in reports of music-induced visual imagery.

In any case, we were able to confirm H2 about how within- and across-participant consistency would differ from each other, supporting Cross' [12] aforementioned idea on the subjectivity of music. Specifically, we show that participants possess greater consistency within themselves across two time-points than consistency with other listeners. Critically, this type of behaviour is not so different to what researchers have described as cross-modal correspondences. Deroy and Spence [21] explain that the sensory connections or cross-modal matching are not only pervasive in everyday life but also remain quite consistent over time, which could be down to environmental regularities or contextual associations.

In a final check, we aimed to re-assess consistency by only including *Storytelling* codes. We viewed this as an important next step to (a) confirm that it was indeed pure visual imagery (i.e., codes where participants report seeing images) that was mostly leading to our observed consistency levels above, and (b) aid comparison with previous work on narrative consistency during music listening [6]. We were able to show that the main conclusions we drew

(regarding comparison of within- and across-participant consistency profiles across code levels) all continued to hold even when looking at this reduced set of codes.

## 5.3. Relating visual imagery consistency to behavioural ratings and individual differences

Next, we explored whether there was a predictive influence of the behavioural measures (prevalence, vividness, music liking, and emotional intensity) and individual differences (general imagery ability, musical training, and participation in the visual arts) on within- and across-participant consistency. We saw links between across-participant consistency and the prevalence and vividness of visual imagery (albeit only L2 for vividness). Interestingly, within-participant consistency was only predicted by vividness and only for L3 codes.

The generally positive relationships found between (across- and within-participant) consistency and prevalence and vividness of visual imagery clearly indicate that the qualitative nature of listeners' visual imagery influences how consistent they are within themselves and with others. While it may not be appropriate to speculate on the nuanced differences seen with regard to levels of consistency, our results suggest that music that is able to induce highly vivid visual imagery tends to produce largely similar content across listeners. Given that we had explicitly chosen to present musical stimuli capable of inducing distinct types of emotion that likely vary in their compositional techniques, we propose that the unique acoustic features of the different excerpts could have influenced how vividly listeners experienced their visual imagery; an idea in line with literature that has highlighted features such as contrast in promoting narrative thinking [4], although is an area of research that is generally in need of much more insight in order to properly ascertain its influence over visual imagery formation [23].

Further, our analyses show that participation in the visual arts does not predict within- or across-participant consistency at either level of granularity, aligning with the idea that those who participate in the visual arts could experience visual imagery at a high enough frequency, vividness, and creative freedom to be inconsistent in content.

## 5.4. Visual imagery, emotion, and aesthetic appeal

In line with previous work which has linked emotion induction with visual imagery formation [24, 25, 29, 35, 36, 51], we saw significant associations between the prevalence and vividness of visual imagery and emotional intensity, whereby the amount of visual imagery and its vividness were positively related to the intensity of emotions felt, confirming H3.

However, our results were not able to speak to the nature of the relationship between emotion and visual imagery. Indeed, there has been increasing debate regarding the directionality of this relationship–recent studies suggest that it is the emotion that is first felt that then leads to visual imagery experience [30], whereas others propose a more complex and mutually beneficial interlink between music-induced visual imagery and emotional induction [29]. In at least one study, it has been shown that manipulating listeners' experience of visual imagery by attempting to hinder it led to a mild suppression of reported induced emotion [35].

Even though our results cannot speak directly to the directionality of the relationship between visual imagery and emotion induction, they extend findings in a number of useful ways. Although our analyses showed that emotional intensity did not predict L3 across-participant consistency, the pattern for L2 consistency was just shy of significance, hinting at the idea that a minor part of what determines high consistency across individuals may be a shared high level of emotion. Future studies could further probe this link by assessing the extent to which emotional valence is a factor driving similarities in visual imagery reports. Secondly, with the

thematic analysis, we obtained a lower-level code dedicated to outlining emotional experiences reported in visual imagery descriptions. Indeed, emotional experiences, whether felt or perceived within the imagery, became a prominent aspect of our categorisations and was highly intertwined in descriptions of different visual imagery scenarios [3]; this being further shown in the fact that it was the second most commonly occurring non-visual subtheme of the *Associations* higher-order theme. Again, while this does not offer evidence into the causal relationship between visual imagery and emotion induction, it speaks to the strong link visual imagery has with emotional experiences.

Finally, in line with previous work that showed visual imagery vividness to be strongly linked with music's aesthetic appeal [26], and in support of H4, we were also able to reflect this finding between prevalence and vividness of visual imagery and music liking responses, whereby liking was positively associated with visual imagery prevalence and vividness. However, this relationship was weaker than seen between visual imagery and emotion, suggesting that it may not be a leading determinant of music-induced visual imagery experience. Here as well, we were unable to provide full insight into the causal relationship between visual imagery and liking. However, the generally weak associations found draw into doubt the idea that aesthetic appeal may play a major role in influencing visual imagery experience. With regard to what may cause a high amount of music-induced visual imagery, one might expect that aspects of acoustic and musical features like melody and harmony [23], structural features like tempo [5], or even interindividual factors like trait empathy [52], may play as important if not more of a role than liking.

## 5.5. The roles of individual differences on music-induced visual imagery

We further explored whether music-induced visual imagery ratings were predicted by generally occurring visual imagery levels. In contrast with previous assessments [35], we found that general visual imagery showed a moderate positive predictive association with music-induced visual imagery, supporting H5 and suggesting that general and music-induced visual imagery may be somewhat interrelated. This finding is partly in line with findings by Küssner and Eerola [3] who showed a positive, albeit small, correlation between imagery vividness and general visual imagery. Nevertheless, due to the inconsistencies found across studies, these results warrant further investigations into the processes that may underly the experience of general as well as music-induced visual imagery.

Furthermore, musical training showed no associations with prevalence and vividness of visual imagery ratings (extending H6 to show that the link is not only weak but in fact non-existent), and is in contrast with past observations that music-induced visual imagery was affected by training due to potential functional benefits [3]. This finding however may not come as such a surprise, as several examples of past literature have similarly found no differences between musicians and non-musicians in their visual imagery abilities [38], instead finding superior involvement of other mental imagery experiences in response to music listening (for example, auditory [53, 54] and kinaesthetic [55, 56]).

Further, in support of H7, we found that those who participate in activities associated with the visual arts provided significantly higher prevalence and vividness of visual imagery ratings, both in terms of aggregated and individual music track responses. This is unsurprising, as it is evident that various artists (visual, musical, etc.) use imagery in their creative processes to stimulate the creation and performance of their art [31]. In the context of music, mental imagery is even found to enhance the perceived creativity of music compositions [32]. Thus, although the general chain of causality is unclear, increased participation within various art modalities (in this case, visual) may increase visual imagery engagement with music.

## 5.6. Implications, limitations, and future directions

With the current research, we have presented a novel methodological approach to probing the content of music-induced visual imagery, a method that we hope will be adopted by future studies seeking to develop the knowledge on the topicality of visual imagery content. The ephemeral nature of visual imagery makes it difficult to measure and draw decisive conclusions. Using our approach of quantifying prevalent themes, we were able corroborate past notions of narrative thinking in the context of music listening and confirm that a large portion of it can be visual in nature.

Our approach may however also suffer from a couple of key limitations; namely that our visual imagery framework was developed in response to only three musical excerpts and were taken from film soundtracks [39]. Our choice of the film genre was partly to ensure that participants were free to provide rich accounts of their visual imagery in response to a programmatic selection of tracks, as well as giving us considerable power with which to analyse consistency, especially when selecting such a low number of listening stimuli. Such decisions mean that our stimuli fall short of being considered entirely comprehensive. It is further a possibility that the predominance of storytelling and media references was an artefact of musical cues present in our chosen tracks that were associated with the development and changes found in film scenes.

However, we propose that these issues could be considered negligible. Margulis [4] found that even when unprompted, individuals listening to instrumental classical music made references to film and television in their narrative open-text reports. Similarities between our study and theirs in revealing a high volume of media references may lie in the fact that our stimuli were predominantly orchestral. Nevertheless, future investigations should aim to incorporate music from genres with contrasting compositional characteristics and instrumentation. One piece of work by Markert and Küssner [57] which utilised ambient music, typically incorporating non-instrumental synthesised compositional techniques, found that while participant reports presented a significant amount of storytelling visual imagery in response to both ambient and classical music excerpts, reports in response to the ambient music tracks also predominantly featured abstract visual imagery (i.e., non-specific images, such as geometric shapes and colours). These findings emphasise the variability in visual imagery that differences in music genre could lead to, and hints at the potential for our framework to be representative of most music genres with further refinement. It would further be relevant to explore differences in the content of our framework and the proportions of themes and consistency levels should listeners be reporting imagery in response to self-selected music. It is likely that this would lead to more personal imagery, due to the higher proportion of autobiographical memories that can be evoked from familiar music [58].

A further important limitation to consider is that the visual imagery content descriptions provided in survey 2 could have been subject to demand characteristics. On the one hand, listeners may have only reported new visual imagery that was not experienced during the first listening instance; on the other, it is possible that some may have felt inclined to report similar visual imagery to that they recalled experiencing in survey 1. Further research on the topic should aim to take this issue into consideration.

Some literature has emphasised the positive effects that eye closure may have on visual imagery experienced in response to music (e.g., [35, 59, 60]), so much so that current imagery-based therapies adopt eye closure as a way to enhance the benefits of rehabilitation [61]. One study that specifically compares the facilitatory effects of eye opening or closure on participants' reported visual imagery experience found that eye closure led to markedly higher visual imagery vividness as well as content [62]. The current design did not offer specific instructions

on whether participants should listen to each musical excerpt with eyes open or closed, thus this action was free to vary across the sample. Given the dramatic influence that the change in instruction can have on visual imagery experience, especially in the context of music listening, such an instruction would be important to include in future research hoping to enhance the experience of visual imagery.

The design of the current study required participants to provide unrestricted reports on the content of their visual imagery to music, implicating their ability to verbalise their visual imagery experiences (i.e., constructing their narrative to music [4]). It is generally well evidenced that music and language are reflected by overlapping electrophysiological correlates [16, 63–67], but some studies identify differences in this regard (e.g., between males and females [68–71], and as a function of musicianship [72–77]). Neuroscientific studies on music-induced visual imagery are only beginning to emerge (see [61, 78] for first evidence of neural signatures). However, we suggest that future studies may seek to combine approaches like those taken in our current paper with emerging insights into neural underpinnings in order to advance knowledge of both the brain and visual imagery during music listening.

What musical characteristics are more likely to result in music-induced visual imagery? While this concept has been minimally addressed, Margulis [4] found that one potential contributing factor in leading listeners to narratively engage with music is musical contrast (sudden and unexpected changes in musical events). Juslin [23] proposes that musical features, such as predictability and repetition, may be driving forces in leading listeners to form visual imagery to music. In any case, the musical features that may link to specific types of visual imagery is, to date, a vast and unanswered question.

Finally, future investigations may consider extending the time between administering the first and second survey to assess the endurance of visual imagery experience more effectively. A yet more comprehensive approach would involve presenting additional administrations of the survey: this is to assess potential modulations more systematically in terms of how levels of within-person consistency change over time.

## 5.7. Conclusion

We have presented a detailed investigation into the visual imagery content that listeners experience in response to music. We show that visual imagery is a highly prevalent aspect of individuals' listening experience, with storytelling being particularly prominent. We also demonstrate the idiosyncrasies of listeners' content consistency by showing that they were, on average, relatively consistent with themselves across timepoints, in contrast to when compared with other listeners.

The ease with which music appears to elicit visual imagery offers further support for the connection between music and language processing with regard to listeners' inclination to derive meaning from the music. We anticipate that our research will set a precedence for further studies to develop and hone our understanding of the inherent visual imagery qualities experienced during music listening.

## Supporting information

**S1 File. Appendices.**
(DOCX)

**S2 File. Supplementary.**
(DOCX)

## Acknowledgments

We would like to thank Olivia Geibel for her assistance with coding the visual imagery description content during the thematic analysis.

## Author Contributions

**Conceptualization:** Sarah Hashim, Lauren Stewart, Mats B. Küssner, Diana Omigie.

**Data curation:** Sarah Hashim.

**Formal analysis:** Sarah Hashim.

**Investigation:** Sarah Hashim.

**Methodology:** Sarah Hashim, Mats B. Küssner, Diana Omigie.

**Supervision:** Mats B. Küssner, Diana Omigie.

**Visualization:** Sarah Hashim.

**Writing – original draft:** Sarah Hashim.

**Writing – review & editing:** Sarah Hashim, Lauren Stewart, Mats B. Küssner, Diana Omigie.

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
