## [Decision Letter · Decision Letter 0]

27 Mar 2023

PONE-D-23-03570Music listening leads to predominantly story-like visual imagery that shows idiosyncratic tendencies

PLOS ONE

Dear Dr. Hashim,

Thank you for submitting your manuscript to PLOS ONE. The manuscript has been evaluated by three very knowledgeable experts in the topical area you are investigating, who all wish to be identified and are Dr. Steffen A. Herff, Dr. Julian Cespedes-Guevara, and Prof. Marta Olivetti Belardinelli.

As you will see when you read their critiques further below and in the attached documents, all three reviewers evaluate positively your work and highlight that it can make a significant contribution to the field. However, the reviewers raise certain concerns that should be addressed in order for the paper to be suitable for publication.

Specifically, all reviewers note that more clarifications should be provided in several parts of the manuscript – most notably, when outlining the formulation of the study aims and hypotheses. Reviewers also make some critical suggestions with respect to statistical analysis and results sections. I believe that addressing these comments would strengthen the clarity of the manuscript and increase its potential impact.

Therefore, I invite you to submit a revised version of the manuscript that addresses the points raised by the reviewers. When revising your manuscript, please consider all issues mentioned in the reviewers’ comments carefully – please outline every change made in response to their comments and provide suitable rebuttals for any comments not addressed. Please note that your revised submission may need to be re-reviewed. 

I look forward to receiving your revised manuscript.

Kind regards,

Ioanna Markostamou, Ph.D.

Academic Editor

PLOS ONE

Journal Requirements:

3. Peer review at PLOS ONE is not double-blinded (https://journals.plos.org/plosone/s/editorial-and-peer-review-process). For this reason, authors should include in the revised manuscript all the information removed for blind review.

Reviewers' comments:

Reviewer's Responses to Questions

**Comments to the Author**

1. Is the manuscript technically sound, and do the data support the conclusions?

Reviewer #1: Yes

Reviewer #2: Yes

Reviewer #3: Yes

2. Has the statistical analysis been performed appropriately and rigorously? 

Reviewer #1: Yes

Reviewer #2: Yes

Reviewer #3: Yes

3. Have the authors made all data underlying the findings in their manuscript fully available?

Reviewer #1: Yes

Reviewer #2: Yes

Reviewer #3: Yes

4. Is the manuscript presented in an intelligible fashion and written in standard English?

Reviewer #1: Yes

Reviewer #2: Yes

Reviewer #3: Yes

5. Review Comments to the Author

**Reviewer #1: **

Dear Dr. Markostamou, Dear Authors,

I had the great pleasure to review the article ‘Music listening leads to predominantly story-like visual imagery that shows idiosyncratic tendencies’ submitted for publication to PLOS ONE (PONE-D-23-03570; review request received: 18.02.2023, review accepted: 20.02.2023, review submitted: 22.02.2023).

This large-scale online study investigates inter- and intra-participant consistency of music-indued visual mental imagery. The manuscript is well written, the experiment is well designed to address the core research questions, and the results fill an important gap in the literature. The manuscript is a great contribution to the field and well suited to the audience of PLOS-One.

I can strongly recommend the manuscript for publication; however, I have a few minor suggestions and three larger concerns regarding formalism and statistical analysis that I recommend the authors should address first. These are detailed in the attached file PONE-D-23-03570 - REVIEW.pdf.

**Reviewer #2: **

I would like to congratulate the author(s) for the rigorous and interesting investigation they carried out. I think it makes a valuable contribution to advancing our knowledge about the phenomenon of music-evoked visual imagery. That said, I have a few suggestions for improving the paper:

(Page 4 Lines 75-82). First, in the introduction, I suggest mentioning Day and Thompson's (2019) work, about how that in many cases, emotional responses to music may occur earlier than visual imagery. This finding suggests that visual imagery is not the necessarily the driving factor behind emotion, but the other way around. The paper does mention this paper in the discussion, but I think it is worth including in the review about the link between imagery and emotion.

(Pages 7 and 8: “The current research”). I found it difficult to understand the aims of the study. I had to read them several times to understand the difference between aims 1 and 4. Additionally, it is not immediately clear what the author(s) mean by “formulating a framework” in aim 2. I suggest rewriting these paragraphs so that their meaning is more readily apparent.

(Pages 8 and 9, Lines 182-204). The author(s) state that they expect to “be able to replicate previous reports of a relationship between visual imagery and both emotion induction and aesthetic appeal” (page 8, lines 182-183). I think this hypothesis needs to be outlined in more detail: what type of relationship did they expect to find? I also suggest labelling the hypotheses stated in this section with numbers and use them throughout the paper to identify them, particularly in the results section.

(Page 10, Lines 223-227). Please provide musical details about the musical excerpts that were used: mode, tempo, instrumentation, style, and the movies from where they were taken from.

(Page 11, Lines 249-250). Did the authors offer any explanation of what synesthesia consists of to the participants?

(Page 15, lines 340-347). I suggest expanding the rationale for using both an across-participants and a between-participants measures of consistency. I did not find this explanation offered by the authors to be sufficient to grasp the usefulness of using both measures: “to allow greater comparison of consistency values within and between individuals, a between-participants consistency was calculated by comparing each participant with a random other individual” (lines 345-347).

(Pages 23-24, lines 841-847; and Pages 26-27, lines 507-510). From my point of view, these sections of the paper are the ones that require revision more urgently, because the author(s) describe the content of the participants’ reported experiences in contradicting terms. For instance, on page 24, lines 485-487, they write: “Whilst not denoting an exact visual image, its presence within reports was considerably prevalent and constituted a prominent attribute of visual imagery experience”. This is confusing, because if the participants did not explicitly report seeing any visual images in their mind, then those experiences do not correspond to “visual experience(s)”. They are indeed, subjective mental experiences while listening to music, but not visual ones. I checked the authors’ original data on the Open Science website, and found that many of the quotes from the participants in the “Feelings & Atmosphere” and “Music” categories correspond to abstract terms such as “Hope and courage", "Sadness", "Haugtiness", "Love, care", "Tranquility peace well-being love", "Fear", "National anthem", "slow music", "A piano like music for study or sleep", "Elevator music", etc. There is nothing in these terms that suggest any visual dimension to them, and therefore, classifying them as “visual imagery” is misleading. From my point of view, these reports correspond to semantic associations about the culturally-shared uses of functions of music.

One important implication of not calling these experiences “visual imagery”, is that the reported percentages of “visual imagery prevalence” should be slightly adjusted.

(Pages 25, lines 507-510). Please provide examples of the terms that participants used in this category.

(Page 35, lines 680-688). I suggest that the authors should emphasize that the fact that a significant proportion of the participants’ reports did not have a “visual” quality to them, implies that by narrowing our focus on researching “visual” experiences we may be missing out an important aspect of listeners’ experiences. They could also mention that this has been found in previous research. For instance, Cespedes-Guevara and Dibben (2022) found that this is a common occurrence even when participants are provided with written narratives about the music meaning. Furthermore, those authors suggest that the power of music to evoke semantic associations may be a crucial factor behind listeners’ mind wandering and emotional experiences while listening to music. That interpretation may also help explain the relatively high levels of consistency found on level 2 in the present investigation.

(Page 42, lines 861-868). I agree with the authors that using movie soundtracks as stimuli was an important limitation of this study. I suggest also mentioning the fact that they only used 3 stimuli was also a limitation.

**Reviewer #3: **

It would be important and interesting to give the distinction between Males and Females for all results presented. If the Authors decide to not perform this distinction, this decision should be indicated as a limitation of the paper.

6. PLOS authors have the option to publish the peer review history of their article (what does this mean?). If published, this will include your full peer review and any attached files.

Reviewer #1: **Yes: **Steffen A. Herff

Reviewer #2: **Yes: **Julian Cespedes-Guevara

Reviewer #3: **Yes: **Marta olivetti Belardinelli

---

## [Author Response · Author response to Decision Letter 0]

28 Jul 2023

Dear Editor,

 Thank you for the opportunity to submit a revised version of our manuscript ‘Music listening leads to predominantly story-like visual imagery that shows idiosyncratic tendencies’ (newly entitled ‘Music listening evokes story-like visual imagery with both idiosyncratic and shared content’) to PLOS ONE. We are grateful to the reviewers for their constructive comments and suggested amendments to our paper. All changes are made on a revised version of the manuscript with tracked changes implemented (entitled ‘Revised Manuscript with Tracked Changes’) as well as in an additional revised unmarked version (entitled ‘Manuscript’), as requested.

Please find below the reviewers’ comments and our responses following each one:

Comments from Reviewer 1 – Dr Steffen A. Herff

Major Comments

Comment: The manuscript in general is of very high quality, however, the results section and statistical approach in its current state is not on par with the rest of the excellent manuscript. Despite the experimental design relying on various related variables that are co-dependent, the authors opted for a very simple analytical approach and base much inference on a large number of individual correlations and t-tests, which (if sticking to the Null-Hypothesis-Significance-Testing analytical frame worked use here) not only leads to a dramatic alpha error accumulation, but also misrepresents the data structure, whilst simultaneously being unnecessarily verbose. This is a shame, as the data set and results are very exciting and constitute a great contribution to the field. A slightly more appropriate analysis that focuses on the main narrative of (in)consistency in music-induced visual mental imagery would substantially increase the quality of the paper, make greater use of the valuable data the authors collected, and streamline the results section. I’ve provided some detailed suggestions, including annotated example code in the detailed comments.

Response: We would like to thank the Reviewer for their perspective on our analytical approach, as well as their detailed annotations at the very end of their document. We agree with the suggested changes to the analysis pipeline and have decided to implement these. These especially include replacing our correlation tests run throughout the Results section with linear mixed effects models that take into consideration the full data across our three pieces.

Comment: The formal calculations of the ‘between’ consistency seem unconventional and possibly a bit off. The authors calculate three metrics of consistency. Within consistency, which is operationalised through Jaccard coefficient index between topic codes of the first and second free format responses, which makes a lot of sense. Across consistency, which is the averaged Jaccard coefficient between a given participant and each other participant in the same condition, which also makes a lot of sense. And a between consistency (largely used for the within vs between consistency comparison), which is the same as the across consistency, only that rather than exhaustively calculates, it samples a single other participant. I see no tangible advantage to the between measure over the across measure, but it does come with a lot of disadvantages (e.g., sample biases, lack of reproducibility, low statistical resolution, and reliance on a particular random seed that would need to be shared). It is not clear why this metric is needed or how it provides ‘greater comparison’ for within vs between tests. I recommend removing the between metric and instead focusing on the across consistency metric, which captures the same information, just much more reliably.

Response: We agree that the distinction between our ‘between’ and ‘across’ measures could have been reflected more convincingly. The initial goal here was to calculate a quasi ‘across’-participant measure that would, in principle, be similar to the within-participant measure (i.e., single-individual comparisons). After further consideration of the issues posed by the Reviewer, we have decided to exclude this measure from the paper and focus solely on the across-participant measure as our metric of inter-individual consistency wherever relevant.

Comment: Comparing consistency levels across levels (i.e., Level 2 vs Leve 3) is formally problematic. This is because the results reported are an innate feature of the data modelling framework. Through the authors’ thematic analysis, the data is structured as a convolutional, embedded hierarchy where each Level 3 codes projects to a point on the Level 2 space. A structure like this necessitates that consistency (as calculated but Jaccard coefficient indices) increases as you move from Level 3 to Level 2, and the factor of increase is largely determined by the distribution of converging coding labels, rather than semantically interpretable properties of participants’ responses. I provide more detail on this in the detailed comments. I strongly recommend removing these analyses from the manuscript, both in the results section as well as their interpretation in the discussion, as this finding is simply an innate feature of your annotation strategy, and instead focusing on the comparison within levels (which are very exciting).

Response: We would like to thank the Reviewer for their explanation into why the analysis between our L2 and L3 codes is problematic (as well as their more detailed example on this issue in another comment later). We agree and, as such, comparisons of the L2 and L3 codes have been removed.

Detailed Comments

Comment: p.1: Title ‘Music listening leads to predominantly story-like visual imagery that shows idiosyncratic tendencies’. – I feel the title might be a bit too strong. You report (even on the detailed Level 3) 20.6% within and 9.8% across consistency. Considering that participants had the opportunity to imagine whatever they wanted, it is remarkable how close your ‘across’ consistency measure got to your ‘within’ consistency measure. A title along the lines of the following might more closely represent your numerical findings: ‘Music listening evokes story-like visual imagery with both idiosyncratic as well as shared content’. Of course, the title is your choice, so please consider this merely a suggestion.

Response: We thank the Reviewer for their suggestion of a more appropriate manuscript title. We have taken this into consideration and have, for the most part, adopted the suggestions proposed by the Reviewer to create a new title: ‘Music listening evokes story-like visual imagery with both idiosyncratic and shared content’

Comment: p.2, l. 31: Abstract: ‘Of the initial sample, 254 respondents completed the survey again three weeks later. – The abstract is very well written and gives an engaging summary of the project. However, this particular sentence seems a bit out of place. At this point of the abstract, the reader does not know yet what the survey is. I recommend moving the sentence to a slightly later place. […]

Response: Agreed. This sentence has now been moved to a later more appropriate place, as suggested by the Reviewer: ‘Further, they completed items assessing a number of individual differences including musical training and general imagery ability. Of the initial sample, 254 respondents completed the survey again three weeks later.’. 

Comment: p.3, l.63: Introduction ‘However, while it is unlikely that music-induced visual imagery solely comprises of static mental pictures, little is known about how it may develop over time.’ – Could just be me, but I found the phrasing here a bit unclear. What does ‘develop over time’ refer to? Based on the content of the paper, I guess it is about test-retest reliability, but since the first part of the sentence is about static images, one might think ‘develop over time’ refers simply to dynamic images within one imagery session. The next sentence then jumps to across listener consistency. Maybe you can rephrase this sentence to emphasize your point.

Response: We agree that this sentence could be clearer. This sentence has now been removed and the paragraph in general has been rephrased to reflect our points more concisely, as follows: ‘For most listeners, forming a narrative in their mind’s eye is a way to engage with heard music [3,4], and such narrative sequences are often reported to be vivid and multi-thematic experiences [5,6]. A few investigations into visual imagery content during music listening have begun to shed light on this idea of imagery consistency across listeners [7] and potential influencing factors [6]. However, the extent to which listeners exhibit similarities in their own visual imagery across listening situations is still an open question.’

Comment: p.3, l. 64: ‘Indeed, to date, there has been no systematic investigation of visual imagery content during music listening and it is unknown the extent to which it is consistent across listeners.’ – I think this might be a bit too strong. There are a few papers that look at across participant consistency of mental imagery in general, and some in music, and I think discussing this might be useful here. For example: […]. It might be worth discussing some of the references above and to be a bit more specific about the exact gap in the literature you are referring to here, currently the statement reads a bit too broad.

Response: We agree with the Reviewer that the phrasing of this sentence is not entirely accurate. Thus, this has been rephased after taking into consideration the suggested references and to reflect the current evidence: ‘A few investigations into visual imagery content during music listening have begun to shed light on this idea of imagery consistency across listeners [7] and potential influencing factors [6].’

Comment: p. 3. L.68-71: “It is also unclear how certain relevant individual differences (such as musical training, general visual imagery ability, synesthetic tendencies or participation in the visual arts) may be associated with prevalence, vividness and consistency of music-induced visual imagery.” – The main motivation statement here reads like ‘we don’t know this yet, so it’s worth investigating’. Personally, I absolutely agree with the sentiment, however, PLOS ONE is a generalist’s journal, so I think it would be important to strengthen the motivation here a bit. From a big picture perspective, what can we learn from exploring this topic in greater detail, what are the implications for our understanding of human cognition, or the role of culture and society that different results would suggest. I think your study is very exciting with many large implications and I think here is the place to highlight a bit further why investigating this topic is important, and why it is worth for the readers of PLOS ONE to read the rest of your exciting paper. Additionally, the unique selling point and main narrative of the paper is about within vs across consistency, so to streamline the narrative, it might be better focus on “It is also unclear how certain relevant individual differences (such as musical training, general visual imagery ability, synesthetic tendencies or participation in the visual arts), imagery vividness and prevalence may be associated with consistency of music-induced visual imagery. (That is focusing on consistency as the main dependent variable of interest, but this is just a suggestion to make your narrative flow nicer)

Response: We agree that the point of this final paragraph could be more convincing. It has therefore now been rephrased to instead emphasise the more pressing aims of the study: ‘For most listeners, forming a narrative in their mind’s eye is a way to engage with heard music [3,4], and such narrative sequences are often reported to be vivid and multi-thematic experiences [5,6]. A few investigations into visual imagery content during music listening have begun to shed light on consistency across listeners [7] and potential influencing factors [6]. However, the extent to which listeners exhibit similarities in their own visual imagery across listening situations is still an open question.’

Comment: p. 4, l. 75: ‘In a similar vein, while visual imagery has been associated with aesthetic and emotional engagement with music, the evidence of such links remains limited.’ – Maybe cite one or two of the theoretical frameworks or recent review articles that conceptualize this in detail here. For example: 

Juslin, P. N. (2013). From everyday emotions to aesthetic emotions: Towards a unified theory of musical emotions. Physics of life reviews, 10(3), 235-266. 

Taruffi, L., & Küssner, M. B. (2022). Visual Mental Imagery, Music, and Emotion. In Music and Mental Imagery.

Response: Agreed. The relevant articles have been cited alongside this statement, as follows: ‘Music-induced visual imagery has been previously associated with aesthetic and emotional engagement with music, but the evidence of such links remains limited [24,25].’

Comment: p. 8, l. 176 “…within, across, and between..’ – The conceptual difference between ‘across’ and ‘between’ is not really clear here. As mentioned earlier and detailed later, the formal difference is also not clear, so I suggest dropping the ‘between’ measure entirely.

Response: We would like to thank the Reviewer for their advice regarding the analysis of this dataset. As mentioned in response to other similar comments, the between-subjects measure has been excluded due to the fact that it does not provide a unique contribution to the results over the across-participant measure. The sentence now reads as follows: ‘iii. To ascertain the extent of the consistency of visual imagery within and across individuals during music listening,’

Comment: p. 9, l. 197 “Further, we predicted we would find little difference between synaesthetes and non-synaesthetes in the consistency of their visual imagery experience; this is due to previous results stating that non-synaesthetes are capable of showing consistency in a colour-picking for letters task, performance on which was predicted by scores in a visual imagery task [39].’ – I find the logical flow here not very clear. You predict no differences between two groups, but use a statistical approach that is not capable of supporting the Null-hypothesis, only rejecting it. If you are interested in supporting the Null-hypothesis, you’ll have to use Bayesian statistics, but frequentist/NHST approaches like the ones you use in your analysis lack the ability to support the null hypothesis (i.e., not rejecting the null is not equal supporting the null). Additionally, synaesthetes is a very broad term, maybe be clear about which precise conditions you are interested in (e.g., all those that have co-evoked percepts involving either sound or visuals or both?), otherwise it isn’t clear why you would consider individuals that say, have co-evoked percepts of pain and smell. Furthermore, why would you predict a lack of an effect between synaesthetes vs non synaesthetes solely based on the finding that non-synaesthetes can do a color-picking task consistency that has some degree of correlation with visual imagery task. I feel like I’m missing something, but currently the logical flow here feels a bit shacky.

Response: We agree with the Reviewer that this hypothesis is not very strongly supported and was not accompanied with the appropriate analyses. While we did collect data on the types of synaesthesia that participants experience, this was significantly varied across this subsample. Thus, this hypothesis, along with any related discussions and analyses, have now been completely removed.

Comment: p.10 l. 223. Materials and Stimuli – Most readers will be unfamiliar with the film-music corpus from Eerola & Vuoskoski and will likely assume that these songs are reasonably familiar to at least some participants. The corpus contains published film music, judged to be relatively unfamiliar by a few raters, but some of the pieces will likely sound familiar to participants (e.g., the Vertigo soundtrack is part of that corpus). I recommend providing a bit more information about the pieces you selected, maybe indicate which exact soundtracks the three selected pieces are from.

Response: We agree that there were key details regarding the musical stimuli missing. These have now been included in the Materials and Stimuli subsection, including information regarding the movies that the excerpts are from and their soundtrack numbers, as follows: ‘Three film music stimuli conveying happy, tender, and fearful emotions were selected from Eerola and Vuoskoski’s [39] database (see, https://www.jyu.fi/hytk/fi/laitokset/mutku/en/research/projects2/past-projects/coe/materials/emotion/soundtracks for access to the original stimuli). These excerpts were obtained from the catalogue of extended 1-min film excerpts (see Appendix of [40], or see, https://www.jyu.fi/hytk/fi/laitokset/mutku/en/research/projects2/past-projects/coe/materials/emotion/soundtracks-1min for access to the original stimuli), validated to be unfamiliar to most listeners and to still convey the intended emotions even in their shorter form. In terms of the films that the tracks were taken from, the excerpt conveying happy emotions was taken from The Untouchables soundtrack (track 6, number 071 from Eerola and Vuoskoski’s set of 110 tracks). The tender excerpt is from the Shine soundtrack (track 10, number 042 from set of 110 tracks). Finally, the fearful excerpt is from the Batman Returns soundtrack (track 5, number 011 from set of 110 tracks). In order to ensure uniformity amongst the musical excerpts, as well as to control the overall length of the survey, all excerpts were edited to last a duration of 45 seconds using Audacity (Version 2.3.2.0). These were also edited to finish with a fade-out to avoid an abrupt ending.’

Comment: p.12, l.265: ‘They were advised to pay attention to any visual imagery that they may be experiencing..’’ – In your instructions you very much highlight visual imagery, which makes a lot of sense given your research questions. However, there is a lot of literature pointing towards the effects of eyes open vs close in visual mental imagery, both in general: […]. Were you participants instructed to keep their eyes open, or to close them? Or was it up to them? I think this is worth mentioning, and maybe discussing in a couple of sentences in the discussion as the simultaneous visual input (or absence of it) can have dramatic effects (the effect are so large, they are used in therapeutic contexts) and this has been explored in particular in the context of – and interaction with- music-induced imagination – the present focus on visual mental imagery warrants a short discussion about this.

Response: The participants did not receive any specific instructions regarding whether they should listen to each musical excerpt with eyes open or closed. We acknowledge that this is an important detail to mention in the manuscript. We have explained this issue, citing a few relevant papers including some suggested by the Reviewer, in the Implications, limitations, and future directions subsection of the Discussion, as follows: ‘Some literature has emphasised the positive effects that eye closure may have on visual imagery experienced in response to music (e.g., [36,64,65]), so much so that current imagery-based therapies adopt eye closure as a way to enhance the benefits of rehabilitation [66]. One study that specifically compares the facilitatory effects of eye opening or closure on participants’ reported visual imagery experience found that eye closure led to markedly higher visual imagery vividness as well as content [67]. The current design did not offer specific instructions on whether participants should listen to each musical excerpt with eyes open or closed, thus this action was free to vary across the sample. Given the dramatic influence that the change in instruction can have on visual imagery experience, especially in the context of music listening, such an instruction would be important to include in future research hoping to enhance the experience of visual imagery.’

Comment: p. 15: Results ‘Across-participants was computed by comparing the list of themes from each individual participant with the lists from every other individual in the sample, resulting in values per participant that were then averaged to create a single consistency value per individual.’ – The general approach is solid, but if you have 352 values per participant (prior to averaging), then this means that you must have averaged across the three pieces. Or did you calculate the Jaccard coefficient index separately for each of the pieces? I think calculating the coefficient separately for each of the three pieces and each participant would make the most sense given your data structure, this would allow you to visualise and test the within vs across-distributions thoroughly and provide very insightful results, however, it isn’t clear from the description which approach was taken.

Response: We agree that the process of calculating the across-participant measure could have been explained more clearly. The coefficient was indeed calculated for each of the three pieces separately, and we did not calculate an across-excerpt average at any stage. The distributions that were analysed and plotted in the Results section later included all of the three pieces, meaning that we had used the full 1,059 (353 values for each excerpt) item dataframe. This explanation has now been amended to make this clearer: ‘We assessed two types of consistency for each musical excerpt separately: within- and across-participant consistency. Within-participant consistency was computed by comparing the lists of themes emerging from each participant’s reports across surveys 1 and 2, and across-participant consistency was computed by comparing the list of themes from each individual participant with the lists from every other individual in the sample, resulting in 352 values per participant for each musical excerpt type (leading to three groups of values) that were then averaged to create a single consistency value per individual per excerpt. With regard to across-participant consistency specifically, individuals were always compared against other individuals’ responses that were within the same excerpt type.’

Comment: p. 15. l, 344: ‘Finally, to allow greater comparison of consistency values within and between individuals, a between-participants consistency was calculated by comparing each participant with a random other individual. With regard to across- and between-participant consistency specifically, individuals were always compared against other individuals’ responses that were within the same excerpt type. – This part really confused me. Based on your description, you are effectively producing a low resolution, sampled metric from your across-participant measure. I do not understand in what way this allows greater comparison. If anything, it allows less reliable comparison, because your comparison would be massively influenced by the particular dyads you are sampling. In addition, this approach is effectively not reproducible unless you share the particular random seed used to pair the dyads, and there is no way of checking whether the dyads were handpicked. Of course, I know that they weren’t, but this approach would theoretically allow it, so this is more a point about general rigor and reportability. I really do not see any advantage in this metric over your across metric. Currently, it feels like you are throwing away a great amount of your precious data without gaining anything in return. Long story short, I strongly suggest ditching the ‘between’ metric and focusing on within vs across. This will provide very compelling support for your project. At the end of the detailed comments section, I’ll propose an alternative structure and analytical approach for your results, you can take as much or little of it on board as you like, but I think it is worth considering.

Response: We thank the Reviewer for their thoughts. As has been mentioned in response to previous comments, we have chosen to exclude the between-participants measure from the manuscript entirely and have taken on board much of the Reviewer’s suggestions regarding the analyses, especially those that enable us to take advantage of the full range of our data.

Comment: p. 17-19 ‘Out of all the ratings of the 353 responds collapsed across the three excepts’ – Why do you collapse across the three pieces? You are throwing away a lot of data. Additionally, the large number of Pearson correlations you are conducting afterwards result in a dramatic alpha error accumulation. If you were to correct for this, basically no result would be significant anymore (I do not suggest you do this). Additionally, based on the degrees of freedom, it seems like all your correlations were also calculated on the aggregated (across excerpts) data even though you deliberately chose those songs to be different across key dimension (such as emotional intensity, or might differ in liking), so you are again throwing away a large amount of data (in the order of magnitude factor 3!). In many instances it isn’t clear why you are correlating particular variables (say liking and emotional intensity) when there wasn’t a clear hypothesis for this. Additionally, the multiple correlations with your target variables ‘e.g., imagery vividness’ using variables that show co-linearity, such as emotional intensity and liking, makes interpreting these results effectively impossible, as all your effects may or may not be tapping into the same variance.

Long story short, I recommend deleting all in-text correlation reporting in the 4.1 section, keep the correlation table for interested readers, and then run a quick model (one for each of your literature guided hypothesis, and ignore those for which you have no literature motivation) for your actual inference that accounts for all these dependencies. This will shorten this section, make the results more interpretable and readable, will make full use of your statistical power, and will be cleaner in terms of avoiding violations of independence. At the end of the detailed comments section, I’ll provide additional detail that might be useful.

Response: We thank the Reviewer for their suggestions on the approaches taken in section 4.1 (now section 4.4). In addition to the aggregated sums and proportions of the prevalence and vividness of visual imagery ratings, we have further included a table outlining the sums and proportions of prevalence and vividness across the three excerpts (see Table 5). Further, we have decided to modify the analytical approach of this subsection, as advised by the Reviewer, and run linear mixed models targeting only our hypothesis-driven relationships.

Comment: p. 26-29. ‘How consistent is visual imagery and to what extent does it depend on abstraction’ – As mentioned above, is see a formal issue in analysing and making claims about increases in consistency between Level 2 and Level 3, and suggest deleting these from the results and discussion. […]

Response: We thank the Reviewer for their comment and also helpful example (not included above for conciseness) regarding why the analysis of abstraction is formally inappropriate. We agree and this change has now been implemented in the Results section of the manuscript, with all later related discussion points removed.

Comment: p. 33. Discussion: As mentioned before, I strongly advise removing all level 3 versus level 2 consistency comparisons, instead, you could just focus on discussing the overall findings that hold even in level 3, which is an interesting pattern of shared and idiosyncratic content. Additionally, if you focus on across vs within and remove the between measurement, then the discussion would also require some adjustment but become much more streamlined in the process.

Response: Having adopted much of the Reviewer’s suggested changes for the study’s analytical approach, we agree that rephrasing certain areas of the Discussion is warranted. These changes have been implemented throughout this section. 

Comment: p. 35, l. 703 ‘At first glance, our finding show notable differences with a recent investigation by Magulis et al. [6] – Agreed, the overall pattern is the same (higher consistency within person/culture, lower consistency across person/culture, with some degree of overlap remaining). The main difference is not in the results, but the metric, one focusing on continuous similarity on a latent variable (cosine similarity) one focus on more discreet topic overlap (yours). Both have their advantages and disadvantages. Cosine similarity allows continuous weighting of similarity, but doesn’t allow exploring distinct topic prevalence or absence. Your approach allows identifying specific themes, but you lose the ability to weight topic importance. Long story short, Margulis measures similarity, you measure whether (amongst others) two reports cover the same topic, regardless of how intense. As a result, I would recommend being a bit more caution with your statement on p. 36, l..714 ‘It was also a more valuable method with which to approach our dataset’. Your method is great and very useful for your research question, but not necessary inherently ‘more valuable’ for the dataset perse.

Response: We agree with the Reviewer that the phrasing of this paragraph is too strong and overshadows the point that was being made, which was a comparison between our own approach and the approach carried out by Margulis and colleagues in answering very similar research questions. This paragraph has now been rephrased to reflect this more appropriately: ‘Approaches used by other studies to address similar questions have also included frequency-based approaches [7] (i.e., summing occurrences of certain content). Critically, the current research aimed to assess consistency with a focus on the presence or absence of particular themes and topics. By allowing analysis of consistency on the basis of themes and topics, our approach provides a way to estimate consistency levels with a focus on particular aspects of content. However, it would be beneficial to consider the advantages of applying a weighted approach (as Margulis and colleagues have done) when scrutinising visual imagery content in combination with the current study’s methods of assessing discrete overlap. Incorporating both topic overlap as well as weighted similarity might provide insight on the significance of specific types of cross-modal content evidently present (as is seen within our thematic framework) in reports of music-induced visual imagery.’

Comment: p. 39, l. 802 “Musical training did however show a very weak but negative significant relationship with music liking, which could suggest that those with training in music may be mildly uninterested in film music.” – This is a bold conclusion in general, but in particular considering that you’ve only tested three songs and averaged across them. I would recommend just deleting this.

Response: We agree with the Reviewer that this sentence should be excluded from the rationale of this analysis.

Comment: p. 3, l. 805: ‘Our results confirmed our expectation that those who experience a form of synaesthesia are no different in the ways they experience prevalence and vividness of visual imagery across music excerpts and in response to individual listening tracks.’ – I would recommend being more careful with your interpretations on synaesthesia. This is a condition with dramatic differences between individuals, you’ve relied on self-report, without additional information about which senses co-create percepts, and your statistical approach (NHST) by definition cannot support the null-hypothesis, only reject it (only a Bayesian Framework can support the Null). So maybe simply rephrase this along the lines of: […]

Response: We agree with the Reviewer that given that we had not probed deeper into the nature of the synaesthesia experienced by participants, our interpretation of this finding should be rephrased. In fact, in line with our response to an earlier related comment, analyses pertaining to synaesthesia have been removed, due to low uniformity of the type of synaesthesia experienced across our sample.

Comment: p. 40, l ‘820’: “Thus, although the directionality is unclear, increased participation within various art modalities (in this case, visual) may increase imagery engagement with music.” – Also a very exciting finding! However, maybe rephrase is slightly, as it is not only a question of directionality (A -> B vs B->A), but it could also be a question an entirely different sets of variables being responsible that affect both (e.g., C -> A & C ->B). Maybe a phrasing like “Thus, although the general chain of causality is unclear, increased participation within various art modalities (in this case, visual) may increase imagery engagement with music. “

Response: We acknowledge the rationale behind the Reviewer’s comment here and agree. We have rephrased this sentence of the Discussion accordingly: ‘Thus, although the general chain of causality is unclear, increased participation within various art modalities (in this case, visual) may increase imagery engagement with music.’

Comment: p. 40, l. 831: ‘The increase in correlation between across-participant consistency and the majority of behavioural ratings and individual differences could point to shared cultural and societal influences, despite our sample being sourced from a wide range of backgrounds.” – I’m not sure I get this, but it might just be me. Which correlations are increasing? Wouldn’t that require an interaction term? And why does the point towards a shared cultural and societal influence? Maybe you can rephrase and elaborate a bit here.

Response: We agree with the Reviewer that this sentence could have been phrased more clearly. However, in the process of updating certain passages of the Discussion due to the extensive changes in the analysis approach, this sentence has now been deleted as it was no longer relevant.

Comment: p. 40 l.83: “The similar levels between within-participants consistency and ratings that we found may indicate…’’ – Same thing, I struggle parsing the sentence. Is this about narrow distributions of all rating scales and the within-participant consistency? I’m not sure, maybe rephrase and elaborate.

Response: We agree with the reviewer that this sentence could have been phrased more clearly. Similarly to the previous comment, this sentence was modified completely to account for the change in results and we believe now expresses our point more clearly, as follows: ‘The generally positive relationships found between (across- and within-participant) consistency and prevalence and vividness of visual imagery clearly indicate that the qualitative nature of listeners’ visual imagery influences how consistent they are within themselves and with others.’

Online Supplement

Comment: These are very useful, thanks a lot! I think the survey1_ratings and survey2_ratings files still contain participants’ prolific IDs in the anon-id column, in addition to the anonymized ‘ID’ column, you might want to remove those.

Response: We thank the Reviewer for highlighting that the data still contained participants’ Prolific IDs. These have now been removed. The numeric anonymous ID index column was retained though, to clarify which participants in survey 1 had returned to complete survey 2.

Suggested Statistical Approach and Results Structure

We would like to thank the Reviewer for their detailed and thorough suggestions for the analytic approach to our manuscript, especially the lines of R code provided, which were extremely useful. We have chosen to implement the approaches suggested in this section as we agree that running these analyses is more effective in utilising the full range of our dataset, and more efficiently helps us to answer our main research questions. The Results section has also been restructured, following the advice of the Reviewer, along with the relevant introductory and discussion points throughout the manuscript to improve the coherence of the overall narrative given the new changes. 

Comments from Reviewer 2 – Dr Julian Cespedes-Guevara

Comment: (Page 4 Lines 75-82). First, in the introduction, I suggest mentioning Day and Thompson's (2019) work, about how that in many cases, emotional responses to music may occur earlier than visual imagery. This finding suggests that visual imagery is not the necessarily the driving factor behind emotion, but the other way around. The paper does mention this paper in the discussion, but I think it is worth including in the review about the link between imagery and emotion.

Response: We agree with the Reviewer that making this addition will strengthen our point. Day and Thompson's (2019) study has now been mentioned in the Introduction, albeit only briefly: this is in order to not compromise the flow of the Introduction (as follows: ‘Thus, associations between music-induced visual imagery and emotion induction are evident, although the directionality of this relationship is still a widely debated topic supported by contrasting evidence [29,30].’), and because we believe that describing the study in detail is better suited as a later discussion point.

Comment: (Pages 7 and 8: “The current research”). I found it difficult to understand the aims of the study. I had to read them several times to understand the difference between aims 1 and 4. Additionally, it is not immediately clear what the author(s) mean by “formulating a framework” in aim 2. I suggest rewriting these paragraphs so that their meaning is more readily apparent.

Response: We would like to thank the Reviewer for highlighting that our aims (specifically aims 1, 2, and 4) could be expressed more clearly. The goal of Aim 1 (now Aim 4) was to address the relationship between visual imagery prevalence and vividness ratings and ratings of music liking and emotional intensity. Whereas the goal of Aim 4 (now Aim 3) was to investigate potential behavioural and individual factors that may be driving how consistent music-induced visual imagery is across time points and individuals. Additionally, the purpose of Aim 2 (now Aim 1) was to run a thematic analysis on our open-text descriptions of visual imagery content, and to organise the patterns found into a multi-layered framework of music-induced visual imagery content. These aims have now been rephrased (and reordered), as follows:

i. ‘To run a thematic analysis to create a hierarchical framework highlighting the prevalent codes and overarching themes found in descriptions of music-induced visual imagery content,

ii. To ascertain the extent of the consistency of visual imagery within and across individuals during music listening,

iii. To examine potential behavioural factors and individual differences (general visual imagery ability, musical training, and participation in the visual arts) that may be driving the rates of within- and across-participant consistency levels,

iv. To test the extent to which the prevalence and vividness of visual imagery is associated with the emotional intensity and aesthetic appeal of music, as well as an array of individual differences (general visual imagery ability, musical training, and participation in the visual arts).’

Comment: (Pages 8 and 9, Lines 182-204). The author(s) state that they expect to “be able to replicate previous reports of a relationship between visual imagery and both emotion induction and aesthetic appeal” (page 8, lines 182-183). I think this hypothesis needs to be outlined in more detail: what type of relationship did they expect to find? I also suggest labelling the hypotheses stated in this section with numbers and use them throughout the paper to identify them, particularly in the results section.

Response: This hypothesis has now been extended to provide a clearer rationale of what is expected: ‘Further, we predicted that we would be able to replicate previous reports of a relationship between visual imagery and both emotion induction [30,35–37] and aesthetic appeal [26]. We specifically predicted that the prevalence and vividness of visual imagery would both be predicted by ratings of emotional intensity (H3) and music liking (H4), in line with previous reports of positive links between these phenomena [23,25,26,35].’

Further, as suggested by the Reviewer, labels for each hypothesis (i.e., H1, H2, etc.) have been implemented where necessary throughout the manuscript.

Comment: (Page 10, Lines 223-227). Please provide musical details about the musical excerpts that were used: mode, tempo, instrumentation, style, and the movies from where they were taken from.

Response: Additional details pertaining to the musical excerpts have now been added, specifically regarding the movies they are from and their soundtrack number. Unfortunately, upon further search, more specific information regarding the mode, tempo, instrumentation, and style of each piece could not be obtained. As a subjective description of these details did not seem appropriate, no further details have been added.

Thus, this has been amended as follows: ‘Three film music stimuli conveying happy, tender, and fearful emotions were selected from Eerola and Vuoskoski’s [39] database (see, https://www.jyu.fi/hytk/fi/laitokset/mutku/en/research/projects2/past-projects/coe/materials/emotion/soundtracks for access to the original stimuli). These excerpts were obtained from the catalogue of extended 1-min film excerpts (see Appendix of [40], or see, https://www.jyu.fi/hytk/fi/laitokset/mutku/en/research/projects2/past-projects/coe/materials/emotion/soundtracks-1min for access to the original stimuli), validated to be unfamiliar to most listeners and to still convey the intended emotions even in their shorter form. In terms of the films that the tracks were taken from, the excerpt conveying happy emotions was taken from The Untouchables soundtrack (track 6, number 071 from Eerola and Vuoskoski’s set of 110 tracks). The tender excerpt is from the Shine soundtrack (track 10, number 042 from set of 110 tracks). Finally, the fearful excerpt is from the Batman Returns soundtrack (track 5, number 011 from set of 110 tracks). In order to ensure uniformity amongst the musical excerpts, as well as to control the overall length of the survey, all excerpts were edited to last a duration of 45 seconds using Audacity (Version 2.3.2.0). These were also edited to finish with a fade-out to avoid an abrupt ending.’

Comment: (Page 11, Lines 249-250). Did the authors offer any explanation of what synesthesia consists of to the participants?

Response: The participants were provided with a brief general explanation on what synaesthesia consists of, illustrated as follows: “Synaesthesia can be defined as an experience relating to one sense or part of the body by the stimulation of another sense or part of the body”. However, analyses and discussion points pertaining to synaesthesia have now been excluded from the manuscript, given that the data collected for this was not extensive enough to draw convincing conclusions. 

Comment: (Page 15, lines 340-347). I suggest expanding the rationale for using both an across-participants and a between-participants measures of consistency. I did not find this explanation offered by the authors to be sufficient to grasp the usefulness of using both measures: “to allow greater comparison of consistency values within and between individuals, a between-participants consistency was calculated by comparing each participant with a random other individual” (lines 345-347).

Response: We would like to thank the Reviewer for highlighting this lack of clarity in the distinction between our across- and between-participant measures. However, it has become clear that the addition of the between-participant measure did not provide a unique contribution to the narrative of the results, over and above the across-participant measure. Thus, the decision was made to exclude this measure from the manuscript entirely.

Comment: (Pages 23-24, lines 841-847; and Pages 26-27, lines 507-510). From my point of view, these sections of the paper are the ones that require revision more urgently, because the author(s) describe the content of the participants’ reported experiences in contradicting terms. For instance, on page 24, lines 485-487, they write: “Whilst not denoting an exact visual image, its presence within reports was considerably prevalent and constituted a prominent attribute of visual imagery experience”. This is confusing, because if the participants did not explicitly report seeing any visual images in their mind, then those experiences do not correspond to “visual experience(s)”. They are indeed, subjective mental experiences while listening to music, but not visual ones. I checked the authors’ original data on the Open Science website, and found that many of the quotes from the participants in the “Feelings & Atmosphere” and “Music” categories correspond to abstract terms such as “Hope and courage", "Sadness", "Haugtiness", "Love, care", "Tranquility peace well-being love", "Fear", "National anthem", "slow music", "A piano like music for study or sleep", "Elevator music", etc. There is nothing in these terms that suggest any visual dimension to them, and therefore, classifying them as “visual imagery” is misleading. From my point of view, these reports correspond to semantic associations about the culturally-shared uses of functions of music.

One important implication of not calling these experiences “visual imagery”, is that the reported percentages of “visual imagery prevalence” should be slightly adjusted.

Response: We appreciate the Reviewer’s comments regarding the consistency and accuracy of how visual imagery is referred to in this section. We did aim to avoid referring to our Associations and References higher order themes as containing pure ‘visual imagery’ and hoped that it would be apparent that these constituted a mixture of ways that participants chose to describe their experience to the music or additional details that participants used to set the scene of their, otherwise, visual experience. This, however, has been clarified in more detail at the beginning of this subsection (as follows, ‘As will be observed below, participant descriptions did not always include experiences that were strictly visual in content and often included descriptions related to affect, musical features, and other forms of mental imagery. This fact has been taken into consideration with regard to analyses and conclusions drawn.’) and the more contradictory statements highlighted by the Reviewer have been either removed or rephrased.

Further, it is not clear whether the percentages that the Reviewer mentions refers to those reported in this subsection, or the percentages of visual imagery prevalence and vividness reported in section 4.1 (now section 4.4). However, assuming that this is in reference to the latter, this percentage is based on the continuous self-report ratings provided by participants on a scale from 1 to 7, and so it is not entirely clear what type of adjustment would be necessary.

Finally, we would like to highlight the additional consistency analyses that we had run using just the Storytelling codes, in order to be certain (in relation to the concerns raised by the Reviewer) that the patterns of differences shown by the models would still stand if the higher order theme containing only visual imagery codes was considered (which they did):

‘Fig D in S2 File presents within- and across-participant distributions using only Storytelling codes, since these, unlike the Associations and References codes, constitute codes that characterise purely visual imagery experiences. We once more show the two distributions to possess distinct shapes and peaks as previously described for all codes. Again, the model comparing Jaccard coefficient values for L2 codes between the within- (Mean = 33.6%, Median = 28.6%) or across-participant (Mean = 21.5%, Median = 21.2%) distributions for Storytelling codes confirmed differences between the two groups (ß = 0.12, SE = 0.01, t = 12.06, p < 0.001). Finally, the model predicting consistency values at the higher granularity L3 codes also demonstrated overall differences between the within- (Mean = 26.1%, Median = 16.6%) and across-participants (Mean = 11.9%, Median = 11.3%) groups (ß = 0.14, SE = 0.01, t = 14.35, p < 0.001).’

Comment: (Pages 25, lines 507-510). Please provide examples of the terms that participants used in this category.

Response: Examples of terms participants used in this subtheme have now been provided immediately after its explanation, as follows: ‘e.g., “As if I were sitting with tea in my hand next to a famous composer like Fryderyk Chopin”, “When the piano started playing I imagined a dark haired male pianist on a black piano…”)’.

Comment: (Page 35, lines 680-688). I suggest that the authors should emphasize that the fact that a significant proportion of the participants’ reports did not have a “visual” quality to them, implies that by narrowing our focus on researching “visual” experiences we may be missing out an important aspect of listeners’ experiences. They could also mention that this has been found in previous research. For instance, Cespedes-Guevara and Dibben (2022) found that this is a common occurrence even when participants are provided with written narratives about the music meaning. Furthermore, those authors suggest that the power of music to evoke semantic associations may be a crucial factor behind listeners’ mind wandering and emotional experiences while listening to music. That interpretation may also help explain the relatively high levels of consistency found on level 2 in the present investigation.

Response: We would like to thank the Reviewer for this suggested addition. We agree that this is a relevant point to make, and it has now been briefly explained in the Discussion, as follows: ‘These findings highlight the considerable prevalence of semantic associations found within listeners’ descriptions in response to the music, even when explicitly instructed to focus their attentions on visual imagery. Such patterns indicate that narrowing one’s research aims on just the visual components of a listeners’ experience could lead to overestimations of its occurrence as well as overshadows its potential unique links to other semantic associations formed in response to the music; one recent study by Cespedes-Guevara and Dibben [50] showed that a considerable portion of listeners’ reports of what went through their minds while listening comprised an array of semantic, personal, as well as visual experiences.’

Comment: (Page 42, lines 861-868). I agree with the authors that using movie soundtracks as stimuli was an important limitation of this study. I suggest also mentioning the fact that they only used 3 stimuli was also a limitation.

Response: We agree with the Reviewer that using only three music stimuli is a key limitation to note. Although this was briefly mentioned, this point has now been emphasised further in the Implications, limitations, and future directions subsection of the Discussion: ‘Our approach may however also suffer from a couple of key limitations; namely that our visual imagery framework was developed in response to only three musical excerpts and were taken from film soundtracks [39]. Our choice of the film genre was partly to ensure that participants were free to provide rich accounts of their visual imagery in response to a programmatic selection of tracks, as well as giving us considerable power with which to analyse consistency, especially when selecting such a low number of listening stimuli. Such decisions mean that our stimuli fall short of being considered entirely comprehensive. It is further a possibility that the predominance of storytelling and media references was an artefact of musical cues present in our chosen tracks that were associated with the development and changes found in film scenes.’

 

Comments from Reviewer 3 – Prof Marta Olivetti Belardinelli

Comment: Lines 210-218: It would be useful to present the different types of participants in a table, also distinguishing Males and Females. This distinction should also be investigated in the results, all the paper along.

Response: We thank the Reviewer for their suggestion. We have now included a table outlining the sums and proportions of the participants’ countries of residence, rather than describing this in the text (please see Table 1). We would further like to thank the Reviewer for their comment regarding the additional investigation of differences between males and females throughout this manuscript. Unfortunately, though, it remains unclear to us why distinguishing by gender is a theoretically necessary and relevant approach for this study. Given it would greatly lengthen an already very long manuscript and given that we have no evidence that this particular question is a pressing issue for pushing forward visual imagery research, we have opted to not address it in this manuscript. Thank you for your understanding. 

Comment: Lines 228-9: the number of the reference should stay after the test name (i.e. after Inventory).

Response: We agree with the Reviewer regarding this change, which has been implemented, as follows: ‘…from Pekala’s Phenomenology of Consciousness Inventory [41].’

Comment: Line 267: a definition of the term “prevalence” is needed.

Response: Agreed. A brief definition of the term “prevalence” (defined as ‘the amount experienced’), as well as the term “vividness” (defined as ‘the clarity with which it was experienced’), has been added.

Comment: Lines 272-3: the order “associated with visual arts,” and synesthesia is the inverse of that indicated in the previous paragraph at lines 250.1.

Response: We thank the Reviewer for highlighting this. However, as a result of the changes made throughout, any analyses and discussion points pertaining to synaesthesia have now been excluded due to low uniformity in the types of synaesthesia present in our dataset.

Comment: Lines 311-3: Please, give more details about the way this hierarchical categorization was performed.

Response: We were a little unclear on what aspects of the hierarchical categorisation the Reviewer thinks require more detail. However, we have included extra information regarding how this process was carried out by the coders, as follows: ‘Commonalities between specific terms were identified from Level 3 (L3) codes, which were sorted and categorised into distinct groups comprising Level 2 (L2) codes. The suitability of the subthemes in this level was reviewed by the two coders, including whether to collapse redundant subthemes or to divide ones that were too diverse. The L2 codes were finally combined to form higher-order Level 1 (L1) themes. The final structure was discussed by the two coders, confirming that it formed an effective hierarchy that offered a parsimonious overview of the content of the free-form descriptions.’

Comment: Line 410: Insert (VA) dopo visual arts and before the coma.

Response: We thank the Reviewer for their suggestion. We agree and have now implemented this change, as follows: ‘In response to the question regarding experience with activities in the visual arts (VA), 20.1% (n = 71) reported that they participate in activities associated with the visual arts, which included activities such as painting, photography, and graphic design. With regard to the averaged excerpt ratings between those who do and do not (NVA) participate in the visual arts, independent samples t-tests showed that those who participate in the visual arts reported significantly more visual imagery prevalence (Mean-VA = 4.63, Mean-NVA = 3.99, t(119.4) = 3.78, p < 0.001) and more vividness (Mean-VA = 4.30, Mean-NVA = 3.74, t(124.1) = 3.37, p < 0.001) than those who do not.’

Comment: Lines 556-7: I think that the indication as first result (as already before in the aims, and afterward line 676) that “visual imagery experience was very prevalent” goes without saying, since according to my comprehension, visual imagery was expressly indicated as the subject’s task. Or were the commitments differently expressed? More interesting is the following indication… 

Lines 686-8:…that visual imagery was accompanied by cross-modal aspects, aesthetic evaluation, and emotions, positively related to images vividness.

Response: We would like to thank the Reviewer for their comments regarding the different sections expressing that music-induced visual imagery was highly prevalent. Participants were indeed instructed prior to listening to each excerpt that they should pay attention to any visual imagery that they may be experiencing throughout listening in preparation for the questions that follow. While on one hand it might go without saying that visual imagery was in fact quite prevalent in our results, on the other, and as has been found in our thematic framework as well as in several past studies [1–4], the experience of music-induced visual imagery is often experienced in combination with a number of non-visual thoughts, and participants often form several semantic, emotional, and personal associations with the music. Thus, we aimed to take this into consideration when phrasing our conclusions regarding the results throughout the Discussion.

References

1. Dahl S, Stella A, Bjørner T. Tell me what you see: An exploratory investigation of visual mental imagery evoked by music. Musicae Scientiae. 2022; 10298649221124862. doi:10.1177/10298649221124862

2. Küssner MB, Eerola T. The content and functions of vivid and soothing visual imagery during music listening: Findings from a survey study. Psychomusicology: Music, Mind, and Brain. 2019;29: 90–99. doi:10.1037/pmu0000238

3. Cespedes-Guevara J, Dibben N. The Role of Embodied Simulation and Visual Imagery in Emotional Contagion with Music. Music & Science. 2022;5: 20592043221093836. doi:10.1177/20592043221093836

4. Margulis EH. An Exploratory Study of Narrative Experiences of Music. MUSIC PERCEPT. 2017;35: 235–248. doi:10.1525/mp.2017.35.2.235

Comment: Lines 856-860: I think that the novelty of the methodological approach should be stressed as an important result of the research besides the…:

Lines 863- 964:… very well indicated key limitations.

Response: We thank the Reviewer for their comment. We do touch upon the novelty of our methodological approach at the start of the Implications, limitations, and future directions subsection. However, we have now emphasised this point further in a few more words: ‘With the current research, we have presented a novel methodological approach to probing the content of music-induced visual imagery, a method that we hope will be adopted by future studies seeking to develop the knowledge on the topicality of visual imagery content.’

Comment: Lines 906-918: Given all that is said above the Conclusion should be rewritten in order to put in evidence the bring-home results.

Response: We agree with the Reviewer that the changes proposed thus far merit an amended Conclusion paragraph. The Conclusion has been rephrased to reflect any changes made to the results, as follows: ‘We have presented a detailed investigation into the visual imagery content that listeners experience in response to music. We show that visual imagery is a highly prevalent aspect of individuals’ listening experience, with storytelling being particularly prominent. We also demonstrate the idiosyncrasies of listeners’ content consistency by showing that they were, on average, relatively consistent with themselves across timepoints, in contrast to when compared with other listeners.

The ease with which music appears to elicit visual imagery offers further support for the connection between music and language processing with regard to listeners’ inclination to derive meaning from the music. We anticipate that our research will set a precedence for further studies to develop and hone our understanding of the inherent visual imagery qualities experienced during music listening.’

 We look forward to hearing from you soon regarding our submission and we are happy to respond to any further questions and comments that you or the reviewers may have.

With best regards, and on behalf of the co-authors,

Sarah Hashim

---

## [Decision Letter · Decision Letter 1]

24 Aug 2023

PONE-D-23-03570R1Music listening evokes story-like visual imagery with both idiosyncratic and shared contentPLOS ONE

Dear Ms Hashim,

Thank you very much for submitting your revised manuscript to PLOS ONE.

I have now received the evaluations of all three of the original reviewers. Please find their comments at the end of this letter. As you will see, the reviewers appreciate very much the changes you have made and point out that their comments and suggestions were well taken into consideration in order to address the issues raised. I also read your revised paper myself and believe that it has now improved substantially.

However, all reviewers have some additional (minor) suggestions for improvement, and I would like to invite you to consider their suggestions in a minor revision of the manuscript.

Please submit your revised manuscript by** Oct 08 2023 11:59PM**. If you will need more time than this to complete your revisions, please reply to this message or contact the journal office at plosone@plos.org. Please include the following items when submitting your revised manuscript:A rebuttal letter that responds to each point raised by the academic editor and reviewer(s). You should upload this letter as a separate file labeled 'Response to Reviewers'.A marked-up copy of your manuscript that highlights changes made to the original version. You should upload this as a separate file labeled 'Revised Manuscript with Track Changes'.An unmarked version of your revised paper without tracked changes. You should upload this as a separate file labeled 'Manuscript'.If applicable, we recommend that you deposit your laboratory protocols in protocols.io to enhance the reproducibility of your results. Protocols.io assigns your protocol its own identifier (DOI) so that it can be cited independently in the future. For instructions see: https://journals.plos.org/plosone/s/submission-guidelines#loc-laboratory-protocols. Additionally, PLOS ONE offers an option for publishing peer-reviewed Lab Protocol articles, which describe protocols hosted on protocols.io. Read more information on sharing protocols at https://plos.org/protocols?utm_medium=editorial-email&utm_source=authorletters&utm_campaign=protocols.

We look forward to receiving your revised manuscript.

Kind regards,

Ioanna Markostamou, Ph.D.

Academic Editor

PLOS ONE

Journal Requirements:

Reviewers' comments:

Reviewer's Responses to Questions

**Comments to the Author**

1. If the authors have adequately addressed your comments raised in a previous round of review and you feel that this manuscript is now acceptable for publication, you may indicate that here to bypass the “Comments to the Author” section, enter your conflict of interest statement in the “Confidential to Editor” section, and submit your "Accept" recommendation.

Reviewer #1: All comments have been addressed

Reviewer #2: All comments have been addressed

Reviewer #3: (No Response)

2. Is the manuscript technically sound, and do the data support the conclusions?

Reviewer #1: Yes

Reviewer #2: Yes

Reviewer #3: Yes

3. Has the statistical analysis been performed appropriately and rigorously? 

Reviewer #1: Yes

Reviewer #2: Yes

Reviewer #3: (No Response)

4. Have the authors made all data underlying the findings in their manuscript fully available?

Reviewer #1: Yes

Reviewer #2: Yes

Reviewer #3: Yes

5. Is the manuscript presented in an intelligible fashion and written in standard English?

Reviewer #1: Yes

Reviewer #2: Yes

Reviewer #3: Yes

6. Review Comments to the Author

Reviewer #1: Dear Dr. Markostamou,

Dear Authors,

I had the great pleasure to review the revised version of the article ‘Music evokes story-like visual imagery with both idiosyncratic and shared content’ submitted for publication to PLOS ONE (PONE-D-23-03570R1; review request received: 01.08.2023, review accepted: 01.08.2023, review submitted: 09.08.2023).

The authors clearly took great care in thoroughly addressing all comments, and I believe the revised version to be an impressive improvement over the last iteration. In particular, all prior comments related to the methodological approach were considered in great detail, including a new statistical approach.

The manuscript is a great contribution to the field and well suited to the audience of PLOS-One.

One final minor suggestion is to include the topic distributions (currently in-text in 4.1.1.-4.13) also in brackets behind the respective topics in Table 2 during the author proof stage. This would provide readers with a very comprehensive understanding of the topic structure when consulting Table 2. But this suggestion is entirely personal preference, and up to the authors to decide.

In summary, I can strongly recommend the manuscript for publication and have no further comments.

Signed,

Steffen Herff

Reviewer #2: I would like to thank the authors for taking into consideration the comments and suggestions I made in the first review to edit the manuscript. I think this new version of the paper represents an improvement to the first one: the paper has now clearer aims, hypotheses and data analyses.

I only have two small suggestions that I think could slightly enhance the readability of the paper:

1) I suggest signalling the hypotheses when reporting each analysis in the results section. For instance, on pages 25, lines 521 onward, the authors could mention that "In order to test hypothesis 2, we asked whether there were differences in consistency when comparing and individual with themselves (...)" etc. Similarly, in the first two paragraph of page 31, they could mention that the models confirm hypotheses 3, 4, 5 and 6, etc.

2) On Page 30, Table 5, I suggest including the aggregated proportions in the table.

These are only suggestions; I leave it to the authors to decide whether incorporating them or not. There’s no need for me to review the paper once more.

Congratulations on your excellent work, I truly think this paper makes an interesting contribution to our understanding of the phenomenon of visual imagery evoked by music.

Reviewer #3: Comments from Reviewer 3 – Prof Marta Olivetti Belardinelli

Comment: Lines 210-218: It would be useful to present the different types of participants in a table, also distinguishing Males and Females. This distinction should also be investigated in the results, all the paper along. Response: We thank the Reviewer for their suggestion. We have now included a table outlining the sums and proportions of the participants’ countries of residence, rather than describing this in the text (please see Table 1). We would further like to thank the Reviewer for their comment regarding the additional investigation of differences between males and females throughout this manuscript. Unfortunately, though, it remains unclear to us why distinguishing by gender is a theoretically necessary and relevant approach for this study. Given it would greatly lengthen an already very long manuscript and given that we have no evidence that this particular question is a pressing issue for pushing forward visual imagery research we have opted to not address it in this manuscript. Thank you for your understanding.

OK for table 1. As regards the gender differences in music and language processing (both implied in the visual imagery task of this research) there is plenty of research assessing it. See for example some of the first studies:

• Koelsch S, Maess B, Grossmann T, Friederici AD. Electric brain responses reveal gender differences in music processing. Neuroreport. 2003 Apr 15;14(5):709-13. doi: 10.1097/00001756-200304150-00010. PMID: 12692468.

• Koelsch S, Grossmann T, Gunter TC, Hahne A, Schröger E, Friederici AD. Children processing music: electric brain responses reveal musical competence and gender differences. J Cogn Neurosci. 2003 Jul 1;15(5):683-93. doi: 10.1162/089892903322307401. PMID: 12965042.

• Koelsch S, Gunter TC, Wittfoth M, Sammler D. Interaction between syntax processing in language and in music: an ERP Study. J Cogn Neurosci. 2005 Oct;17(10):1565-77. doi: 10.1162/089892905774597290. PMID: 16269097.

• Koelsch S. Music-syntactic processing and auditory memory: similarities and differences between ERAN and MMN. Psychophysiology. 2009 Jan;46(1):179-90. doi: 10.1111/j.1469-8986.2008.00752.x. Epub 2008 Nov 21. PMID: 19055508.

• Carrus E, Pearce MT, Bhattacharya J. Melodic pitch expectation interacts with neural responses to syntactic but not semantic violations. Cortex. 2013 Sep;49(8):2186-200. doi: 10.1016/j.cortex.2012.08.024. Epub 2012 Sep 20. PMID: 23141867.

According to my meaning, it is therefore important to cite the topic as a limitation of the study tied to the manuscript length, or as a feature research goal, as you prefer.

Comment: Lines 272-3: the order “associated with visual arts,” and synesthesia is the inverse of that indicated in the previous paragraph at lines 250.1. Response: We thank the Reviewer for highlighting this. However, as a result of the changes made throughout, any analyses and discussion points pertaining to synaesthesia have now been excluded due to low uniformity in the types of synaesthesia present in our dataset. OK

Comment: Lines 311-3: Please, give more details about the way this hierarchical categorization was performed. Response: We were a little unclear on what aspects of the hierarchical categorisation the Reviewer thinks require more detail. However, we have included extra information regarding how this process was carried out by the coders, as follows: ‘Commonalities between specific terms were identified from Level 3 (L3) codes, which were sorted and categorised into distinct groups comprising Level 2 (L2) codes. The suitability of the subthemes in this level was reviewed by the two coders, including whether to collapse redundant subthemes or to divide ones that were too diverse. The L2 codes were finally combined to form higher-order Level 1 (L1) themes. The final structure was discussed by the two coders, confirming that it formed an effective hierarchy that offered a parsimonious overview of the content of the free-form descriptions.’ 27

Comment: Line 410: Insert (VA) dopo visual arts and before the coma. Response: We thank the Reviewer for their suggestion. We agree and have now implemented this change, as follows: ‘In response to the question regarding experience with activities in the visual arts (VA), 20.1% (n = 71) reported that they participate in activities associated with the visual arts, which included activities such as painting, photography, and graphic design. With regard to the averaged excerpt ratings between those who do and do not (NVA) participate in the visual arts, independent samples t-tests showed that those who participate in the visual arts reported significantly more visual imagery prevalence (Mean-VA = 4.63, Mean-NVA = 3.99, t(119.4) = 3.78, p < 0.001) and more vividness (Mean-VA = 4.30, Mean-NVA = 3.74, t(124.1) = 3.37, p < 0.001) than those who do not.’ Good precision.

Comment: Lines 556-7: I think that the indication as first result (as already before in the aims, and afterward line 676) that “visual imagery experience was very prevalent” goes without saying, since according to my comprehension, visual imagery was expressly indicated as the subject’s task. Or were the commitments differently expressed? More interesting is the following indication… Lines 686-8:…that visual imagery was accompanied by cross-modal aspects, aesthetic evaluation, and emotions, positively related to images vividness. Response: We would like to thank the Reviewer for their comments regarding the different sections expressing that music-induced visual imagery was highly prevalent. Participants were indeed instructed prior to listening to each excerpt that they should pay attention to any visual imagery that they may be experiencing throughout listening in preparation for the questions that follow. While on one hand it might go without saying that visual imagery was in fact quite prevalent in our results, on the other, and as has been found in our thematic framework as well as in several past studies [1–4], the experience of music-induced visual imagery is often 28 experienced in combination with a number of non-visual thoughts, and participants often form several semantic, emotional, and personal associations with the music. Thus, we aimed to take this into consideration when phrasing our conclusions regarding the results throughout the Discussion. References 1. Dahl S, Stella A, Bjørner T. Tell me what you see: An exploratory investigation of visual mental imagery evoked by music. Musicae Scientiae. 2022; 10298649221124862. doi:10.1177/10298649221124862 2. Küssner MB, Eerola T. The content and functions of vivid and soothing visual imagery during music listening: Findings from a survey study. Psychomusicology: Music, Mind, and Brain. 2019;29: 90–99. doi:10.1037/pmu0000238 3. Cespedes-Guevara J, Dibben N. The Role of Embodied Simulation and Visual Imagery in Emotional Contagion with Music. Music & Science. 2022;5: 20592043221093836. doi:10.1177/20592043221093836 4. Margulis EH. An Exploratory Study of Narrative Experiences of Music. MUSIC PERCEPT. 2017;35: 235–248. doi:10.1525/mp.2017.35.2.235 It is exactly what I asked to stress.

Comment: Lines 856-860: I think that the novelty of the methodological approach should be stressed as an important result of the research besides the…: Lines 863- 964:… very well indicated key limitations. Response: We thank the Reviewer for their comment. We do touch upon the novelty of our methodological approach at the start of the Implications, limitations, and future directions subsection. However, we have now emphasised this point further in a few more words: ‘With the current research, we have presented a novel methodological approach to probing the content of music-induced visual imagery, a method that we hope will be adopted by future studies seeking to develop the knowledge on the topicality of visual imagery content.’ Great job!

7. PLOS authors have the option to publish the peer review history of their article (what does this mean?). If published, this will include your full peer review and any attached files.

Reviewer #1: **Yes: **Steffen A. Herff

Reviewer #2: **Yes: **Julian Cespedes-Guevara

Reviewer #3: **Yes: **Marta Olivetti Belardinelli

---

## [Author Response · Author response to Decision Letter 1]

18 Sep 2023

Dear Editor,

 Thank you for the opportunity to submit a revised version of our manuscript ‘Music listening evokes story-like visual imagery with both idiosyncratic and shared content’ to PLOS ONE. We are grateful to the reviewers for their constructive comments and suggested amendments to our paper. All changes are made on a revised version of the manuscript with tracked changes implemented (entitled ‘Revised Manuscript with Tracked Changes’) as well as in an additional revised unmarked version (entitled ‘Manuscript’), as requested.

Please find below the reviewers’ comments and our responses following each one:

Comments from Reviewer 1 – Dr Steffen A. Herff

Comment: One final minor suggestion is to include the topic distributions (currently in-text in 4.1.1.-4.13) also in brackets behind the respective topics in Table 2 during the author proof stage. This would provide readers with a very comprehensive understanding of the topic structure when consulting Table 2. But this suggestion is entirely personal preference, and up to the authors to decide.

Response: We thank the reviewer for their suggestion. The L1 and L2 topic distributions, originally only presented in the main text, have now also been included in Table 2 next to their corresponding theme label.

Comments from Reviewer 2 – Dr Julian Cespedes-Guevara

I only have two small suggestions that I think could slightly enhance the readability of the paper:

Comment: 1) I suggest signalling the hypotheses when reporting each analysis in the results section. For instance, on pages 25, lines 521 onward, the authors could mention that "In order to test hypothesis 2, we asked whether there were differences in consistency when comparing and individual with themselves (...)" etc. Similarly, in the first two paragraph of page 31, they could mention that the models confirm hypotheses 3, 4, 5 and 6, etc.

Response: We agree with the reviewer that signalling the individual hypotheses across the Results section would provide more clarity. This has been implemented throughout this section as suggested.

Comment: 2) On Page 30, Table 5, I suggest including the aggregated proportions in the table. These are only suggestions; I leave it to the authors to decide whether incorporating them or not. There’s no need for me to review the paper once more.

Response: We agree with this change and have included the aggregated values within the table alongside the individual musical excerpt values (see Table 5). Accordingly, the paragraph preceding this table that originally described the aggregated proportions has been rephrased to describe the table as a whole instead, as follows: ‘Table 5 presents a summary of visual imagery prevalence and vividness ratings divided and aggregated by musical excerpt. These values demonstrate high proportions of visual imagery prevalence and vividness across all musical excerpts, that persist even when one considers the individual excerpt types. Higher prevalence and vividness levels of visual imagery can be seen in response to the Fearful excerpt than the Happy and Tender excerpts, which both exhibit almost equal proportions.’. 

Comments from Reviewer 3 – Prof Marta Olivetti Belardinelli

Comment: As regards the gender differences in music and language processing (both implied in the visual imagery task of this research) there is plenty of research assessing it. See for example some of the first studies:

• Koelsch S, Maess B, Grossmann T, Friederici AD. Electric brain responses reveal gender differences in music processing. Neuroreport. 2003 Apr 15;14(5):709-13. doi: 10.1097/00001756-200304150-00010. PMID: 12692468.

• Koelsch S, Grossmann T, Gunter TC, Hahne A, Schröger E, Friederici AD. Children processing music: electric brain responses reveal musical competence and gender differences. J Cogn Neurosci. 2003 Jul 1;15(5):683-93. doi: 10.1162/089892903322307401. PMID: 12965042.

• Koelsch S, Gunter TC, Wittfoth M, Sammler D. Interaction between syntax processing in language and in music: an ERP Study. J Cogn Neurosci. 2005 Oct;17(10):1565-77. doi: 10.1162/089892905774597290. PMID: 16269097.

• Koelsch S. Music-syntactic processing and auditory memory: similarities and differences between ERAN and MMN. Psychophysiology. 2009 Jan;46(1):179-90. doi: 10.1111/j.1469-8986.2008.00752.x. Epub 2008 Nov 21. PMID: 19055508.

• Carrus E, Pearce MT, Bhattacharya J. Melodic pitch expectation interacts with neural responses to syntactic but not semantic violations. Cortex. 2013 Sep;49(8):2186-200. doi: 10.1016/j.cortex.2012.08.024. Epub 2012 Sep 20. PMID: 23141867.

According to my meaning, it is therefore important to cite the topic as a limitation of the study tied to the manuscript length, or as a feature research goal, as you prefer.

Response: We thank the reviewer for their suggestion and useful resources. We have now included a paragraph in the Implications, limitations, and future directions subsection briefly addressing this, including additional references to clarify our points, as follows:

‘The design of the current study required participants to provide unrestricted reports on the content of their visual imagery to music, implicating their ability to verbalise their visual imagery experiences (i.e., constructing their narrative to music [4]). It is generally well evidenced that music and language are reflected by overlapping electrophysiological correlates [16,63–67], but some studies identify differences in this regard (e.g., between males and females [68–71], and as a function of musicianship [72–77]). Neuroscientific studies on music-induced visual imagery are only beginning to emerge (see [61,78] for first evidence of neural signatures). However, we suggest that future studies may seek to combine approaches like those taken in our current paper with emerging insights into neural underpinnings in order to advance knowledge of both the brain and visual imagery during music listening.’

 We look forward to hearing from you soon regarding our submission and we are happy to respond to any further questions and comments that you or the reviewers may have.

With best regards, and on behalf of the co-authors,

Sarah Hashim

---

## [Decision Letter · Decision Letter 2]

12 Oct 2023

Music listening evokes story-like visual imagery with both idiosyncratic and shared content

PONE-D-23-03570R2

Dear Dr. Hashim,

Thank you for aking into consideration all the comments and suggestions made by the reviewers and submiting your revised manuscript at PLOS ONE.

I have now received the reviews of the three original reviewers, and as you will see from their comments at the end of this letter, all reviewers evaluated very positively your paper in its current form and did not raise any further points of critique.

Therefore, I am happy to inform you that your manuscript has been judged scientifically suitable for publication at PLOS ONE and will be formally accepted for publication once it meets all outstanding technical requirements.

Kind regards,

Ioanna Markostamou, Ph.D.

Academic Editor

PLOS ONE

Reviewers' comments:

Reviewer's Responses to Questions

**Comments to the Author**

1. If the authors have adequately addressed your comments raised in a previous round of review and you feel that this manuscript is now acceptable for publication, you may indicate that here to bypass the “Comments to the Author” section, enter your conflict of interest statement in the “Confidential to Editor” section, and submit your "Accept" recommendation.

Reviewer #1: All comments have been addressed

Reviewer #2: All comments have been addressed

Reviewer #3: All comments have been addressed

2. Is the manuscript technically sound, and do the data support the conclusions?

Reviewer #1: Yes

Reviewer #2: Yes

Reviewer #3: Yes

3. Has the statistical analysis been performed appropriately and rigorously? 

Reviewer #1: Yes

Reviewer #2: Yes

Reviewer #3: Yes

4. Have the authors made all data underlying the findings in their manuscript fully available?

Reviewer #1: Yes

Reviewer #2: Yes

Reviewer #3: Yes

5. Is the manuscript presented in an intelligible fashion and written in standard English?

Reviewer #1: Yes

Reviewer #2: Yes

Reviewer #3: Yes

6. Review Comments to the Author

Reviewer #1: (No Response)

Reviewer #2: I would like to thank the authors for taking into consideration the comments and suggestions I made, and I congratulate them for carrying out a piece of research that advances our understanding of the music-evoked visual imagery phenomenon.

Reviewer #3: My comments were satisfied and the paper sounds clearly ready to be published. I support the publication of this paper

7. PLOS authors have the option to publish the peer review history of their article (what does this mean?). If published, this will include your full peer review and any attached files.

Reviewer #1: **Yes: **Dr. Steffen A. Herff

Reviewer #2: **Yes: **Julian Cespedes-Guevara,PhD

Reviewer #3: **Yes: **Marta Olivetti Belardinelli

---

## [Editor Report · Acceptance letter]

17 Oct 2023

PONE-D-23-03570R2 

Music listening evokes story-like visual imagery with both idiosyncratic and shared content 

Dear Dr. Hashim:

I'm pleased to inform you that your manuscript has been deemed suitable for publication in PLOS ONE. Congratulations! Your manuscript is now with our production department. 

Kind regards, 

on behalf of

Dr. Ioanna Markostamou 

Academic Editor

PLOS ONE